

# Systematic review: progress in EEG-based speech imagery brain-computer interface decoding and encoding research

Ke Su[1] and Liang Tian[2]

[1] School of Art and Design, Qilu University of Technology (Shandong Academy of Sciences), Jinan, Shandong, China
[2] Department of Computer Science and Technology, Qilu University of Technology (Shandong Academy of Sciences), Jinan, Shandong, China

## ABSTRACT

This article systematically reviews the latest developments in electroencephalogram (EEG)-based speech imagery brain-computer interface (SI-BCI). It explores the brain connectivity of SI-BCI and reveals its key role in neural encoding and decoding. It analyzes the research progress on vowel-vowel and vowel-consonant combinations, as well as Chinese characters, words, and long-words speech imagery paradigms. In the neural encoding section, the preprocessing and feature extraction techniques for EEG signals are discussed in detail. The neural decoding section offers an in-depth analysis of the applications and performance of machine learning and deep learning algorithms. Finally, the challenges faced by current research are summarized, and future directions are outlined. The review highlights that future research should focus on brain region mechanisms, paradigms innovation, and the optimization of decoding algorithms to promote the practical application of SI-BCI technology.

## INTRODUCTION

A brain-computer interface (BCI) refers to a technology that establishes a direct connection between the human brain and external devices. By detecting and decoding brain activity, human intentions can be transmitted directly to these devices, enabling communication between the brain and external equipment (*Wolpaw, Millan & Ramsey, 2020*). Based on the need for external stimulation, BCIs are primarily categorized into evoked neurocognitive paradigms and spontaneous neurocognitive paradigms. This study focuses on one type of spontaneous psychological paradigm—speech imagery. Speech imagery refers to mentally articulating words without actual vocalization or facial movement (*Liu et al., 2022*). This phenomenon engages neural mechanisms related to cognition, memory, learning, and thought processes. The integration of these neural mechanisms with BCI technology forms a speech imagery-based BCI, which detects neural signals associated with speech processes through external recording devices and decodes the brain's speech intentions (which have been encoded into neural signals). These

Corresponding author
Ke Su, coco_su0716@163.com

decoded signals can then be used to generate commands for controlling external devices, thus facilitating interaction with external systems (*Lee, Lee & Lee, 2020*; *Pawar & Dhage, 2022*; *Qureshi et al., 2017*).

In recent years, speech imagery-based brain-computer interface (SI-BCI) has emerged as a prominent research focus, with numerous researchers conducting extensive studies from various perspectives. Generally, these studies can be categorized into four main areas: the neural mechanisms of brain regions during speech imagery, the design of speech imagery paradigms, speech imagery encoding techniques, and speech imagery decoding techniques. Among these, the neural mechanisms of brain regions constitute the physiological foundation for speech imagery. The speech imagery paradigm is an experimental task designed based on these neural mechanisms, while speech imagery encoding and decoding refer to technological processes that facilitate the translation of human intentions into commands within this paradigm.

Currently, research on the neural mechanisms of brain regions primarily focuses on analyzing the overlap in activation patterns across different brain regions during between actual speech and speech imagery states using various brain signal acquisition techniques. *Lu et al. (2023)* employed functional magnetic resonance imaging (fMRI) to compare the neural mechanisms of actual speech perception and speech imagery. They found that both states shared activation in the bilateral superior temporal gyrus (STG) and supplementary motor area (SMA). However, speech imagery specifically enhanced activity in the left inferior frontal gyrus (Broca's area), revealing the neural basis of the motor-simulation-dependent neural basis of internal language generation. *Beyeler et al. (2016)* applied optogenetic labeling combined with electrophysiological recording techniques to dissect the encoding differences in neuronal populations projecting from the basolateral amygdala (BLA) to the nucleus accumbens (NAc), central amygdala (CeA), and ventral hippocampus (vHPC) during the retrieval of positive and negative valence memories. They discovered that BLA-NAc projections preferentially encode positive valence cues, BLA-CeA projections favor negative valence cues, and BLA-vHPC projections exhibit balanced responses to both valences, with more pronounced activation during speech imagery. These findings offer foundational scientific insights for future research on neural decoding of speech imagery. *Peña Serrano, Jaimes-Reátegui & Pisarchik (2024)* constructed an event-related coherence-based magnetoencephalography (MEG) hypergraph network analysis technique to reveal the dynamic patterns of functional connectivity across different brain regions (frontal, parietal, temporal, and occipital lobes) at various frequency bands (delta, theta, alpha, beta, and gamma) during different perceptual tasks. Their study found differences in activation in the left inferior frontal region between spontaneous and evoked speech imagery, with task-evoked speech imagery significantly enhancing activation in this region, while spontaneous speech imagery exhibited relatively reduced activation (*Hurlburt et al., 2016*). *Rekrut, Selim & Krüger (2022)* designed a novel training method that successfully applied transfer learning to train a silent speech classifier on overt speech EEG data and subsequently applied it to covert speech data, demonstrating the feasibility of transfer learning in silent speech BCIs. From these prior studies, it is evident that while there are connections between speech imagery and actual speech, applying

conclusions derived from actual speech directly to speech imagery remains a subject of debate. The activation states of brain regions during the processing of different types of speech vary, necessitating more detailed regional brain analysis.

Research trends in speech imagery paradigm design have evolved from initial focuses on vowels and vowel–consonant combinations to more complex units such as words, Chinese characters, and potentially short sentences. For instance, *DaSalla et al. (2009)* and *Idrees & Farooq (2016a)* developed paradigms centered around the vowels /a/ and /u/. Meanwhile, *Wang et al. (2021)* and *Huang et al. (2022)* have conducted extensive, long-term studies on paradigms involving Chinese characters such as "左(left)", "移(move)", "壹(one)", "右 (right)". *Liu et al. (2022)* offered a comprehensive review of experimental paradigms, providing in-depth analyses of various design strategies. However, their discussion of signal decoding techniques lacked depth and comprehensive analysis.

In the field of speech imagery encoding, research has primarily focused on artifact removal during signal acquisition and algorithmic enhancements in feature extraction. *Chen & Pan (2020)* investigated the application of speech imagery in brain–computer interaction, emphasizing innovations in signal acquisition and processing methods. *Schultz et al. (2017)* described various physiological signals involved in speech production and their recording techniques, offering valuable insights into the biological basis of speech. *Cooney, Folli & Coyle (2018)* focused on the physiological foundations of speech, providing a detailed analysis of speech production processes. However, these studies primarily addressed physiological mechanisms and signal acquisition and encoding, while lacking systematic reviews of signal decoding methods and experimental paradigms. *Alzahrani, Banjar & Mirza (2024)* focused on classifying command-based vocabulary (*e.g.*, "up," "down," "left," "right") for BCIs, examining the effects of feature extraction techniques—including deep learning, adaptive optimization, and frequency-specific decomposition—on classification performance. They compared traditional machine learning techniques with deep learning approaches and discussed the influence of brain lateralization in imagined speech. *Lopez-Bernal et al. (2022)* reviewed relevant literature published since 2009, emphasizing electroencephalogram (EEG) signal preprocessing, feature extraction, and classification techniques. These two studies offer more comprehensive comparisons between traditional machine learning methods and deep learning approaches.

Speech imagery decoding has long been a central focus in the development of SI-BCI systems. Recent trends in this field emphasize the use of machine learning and deep learning algorithms to enhance decoding accuracy. *Rahman et al. (2024)* conducted a comprehensive comparison of classification algorithms, including support vector machines (SVM), random forests (RF), and various deep learning techniques. *Zhang et al. (2024)* analyzed multiple deep learning frameworks and, in contrast to earlier work, provided an in-depth examination of the EEG datasets employed in various studies. These efforts underscore ongoing progress in utilizing advanced computational methods to improve the precision and reliability of speech imagery decoding in SI-BCI systems.

Despite existing reviews covering the aforementioned four areas, a significant gap persists—namely, the absence of a comprehensive and systematic analysis of recent

advancements in BCI paradigm definition and classification, SI-BCI neural encoding, and neural decoding, all grounded in the neural mechanisms of speech imagery. *Panachakel & Ramakrishnan (2021)* presented a 2021 review summarizing various methods for decoding imagined speech from EEG signals developed over the past decade. Their review covered key aspects such as data acquisition, signal processing, feature extraction, and classification methods, while also offering preliminary insights into the relationship between speech imagery and brain regions. However, with the rapid evolution of technology and theory, their study has become outdated and does not meet the requirements of current research. In addition, we have assessed the advantages and limitations of various signal acquisition methods—including EEG, MEG, functional near-infrared spectroscopy (fNIRS), and fMRI—based on practicality, research volume, cost, and generalizability. We conclude that EEG is currently the most suitable approach and have accordingly identified EEG signal acquisition as a key prerequisite for this study (*Panachakel & Ramakrishnan, 2021*; *Kaongoen et al., 2023*).

This review aims to explore recent advancements in EEG-based SI-BCI encoding and decoding by providing a systematic and comprehensive overview. It focuses on dissecting SI-BCI paradigms, neural encoding, and neural decoding, with particular attention to existing challenges, emerging trends, and future prospects (the data collection process is shown in Fig. 1). First, brain region connectivity is discussed as the physiological foundation of SI-BCI systems. Next, existing SI-BCI paradigms are analyzed based on the characteristics of brain region connectivity, followed by a detailed exposition of relevant neural encoding and decoding methods. Finally, key challenges and potential developmental directions are summarized. The key research contributions of this review are outlined as follows.

1. To examine brain region connectivity in SI-BCI, establishing a physiological foundation for the development of paradigms, neural encoding, and decoding techniques.

2. To clarify the relationship between SI-BCI paradigms and neural encoding/decoding, and to elaborate on recent advancements across different paradigms.

3. To explain neural encoding in SI-BCI, including EEG preprocessing and feature extraction techniques.

4. To present a detailed discussion on neural decoding, including available datasets, participant numbers, applied machine learning and deep learning algorithms, and a comprehensive model evaluation.

5. To identify challenges in SI-BCI—including those related to signal acquisition quality, brain region connectivity, and neural encoding—while highlighting future directions such as word and sentence-level decoding and paradigm integration.

The structure of this review is outlined as follows (Fig. 2). "Introduction" provides an introduction to the background of the problem, outlines the motivation for the study, and emphasizes the distinctive features of this article. "Connectivity of Brain Regions In SI-BCI" details the literature search strategy and specifies the inclusion criteria. "SI-BCI Paradigm" systematically explores brain region connectivity in SI-BCI (*He et al., 2019*). "SI-BCI Neural Encoding" defines SI-BCI paradigms and discusses various types of paradigm structures. "SI-BCI Neural Decoding" analyzes neural encoding techniques,

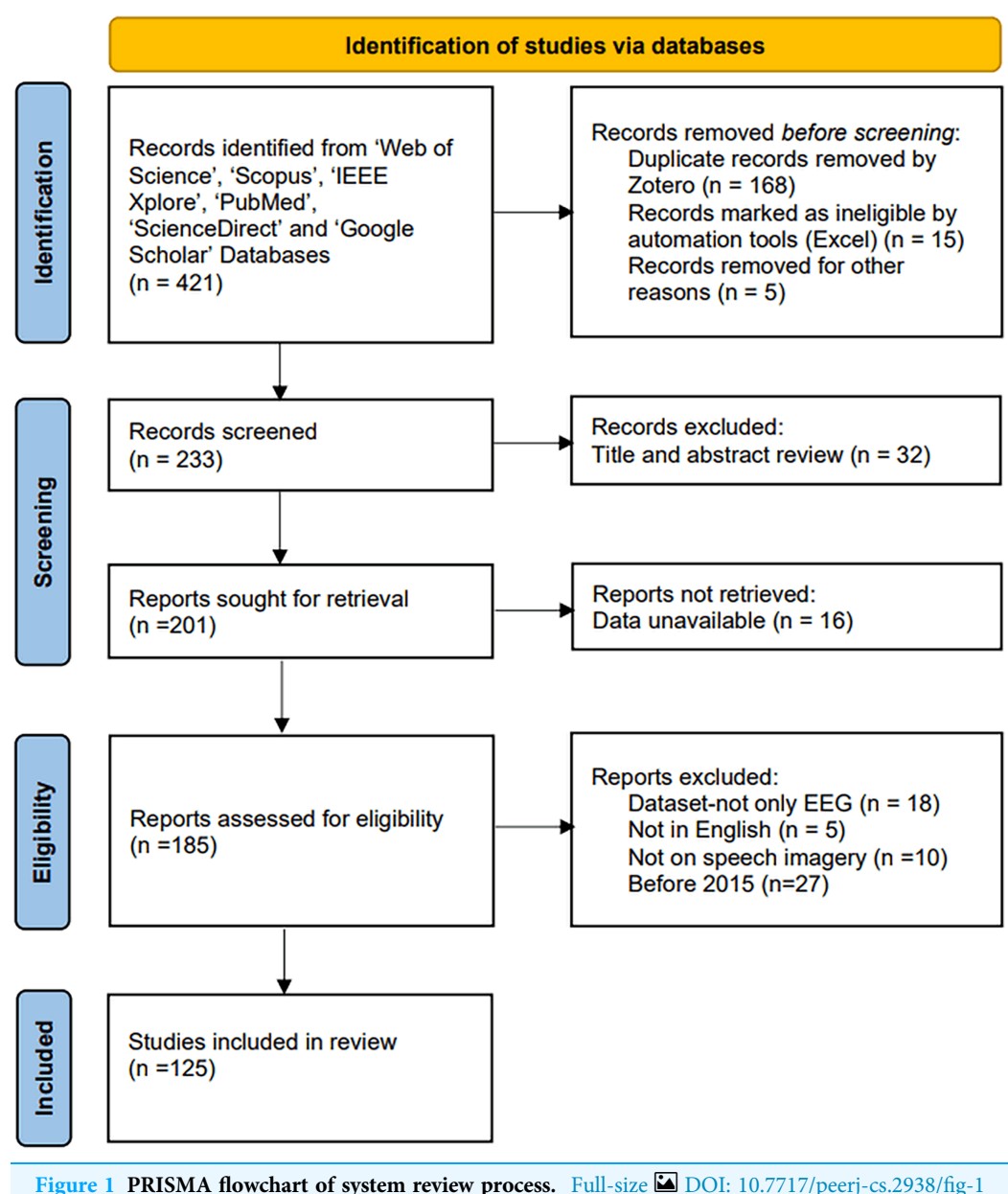

**Figure 1 PRISMA flowchart of system review process.**

focusing on EEG signal preprocessing and feature extraction. "Challenges and Prospects" offers a comprehensive discussion of neural decoding techniques (*Kristensen, Subhi & Puthusserypady, 2020*), including datasets, participant numbers, machine learning (ML) and deep learning (DL) algorithms, and a model performance evaluation table. "Conclusions" addresses current challenges in the field and explores potential future research directions.

## Survey methodology

This review utilized a method called PRISMA (Preferred Reporting Items for Systematic Reviews and Meta-Analyses), which is a protocol for conducting systematic reviews and
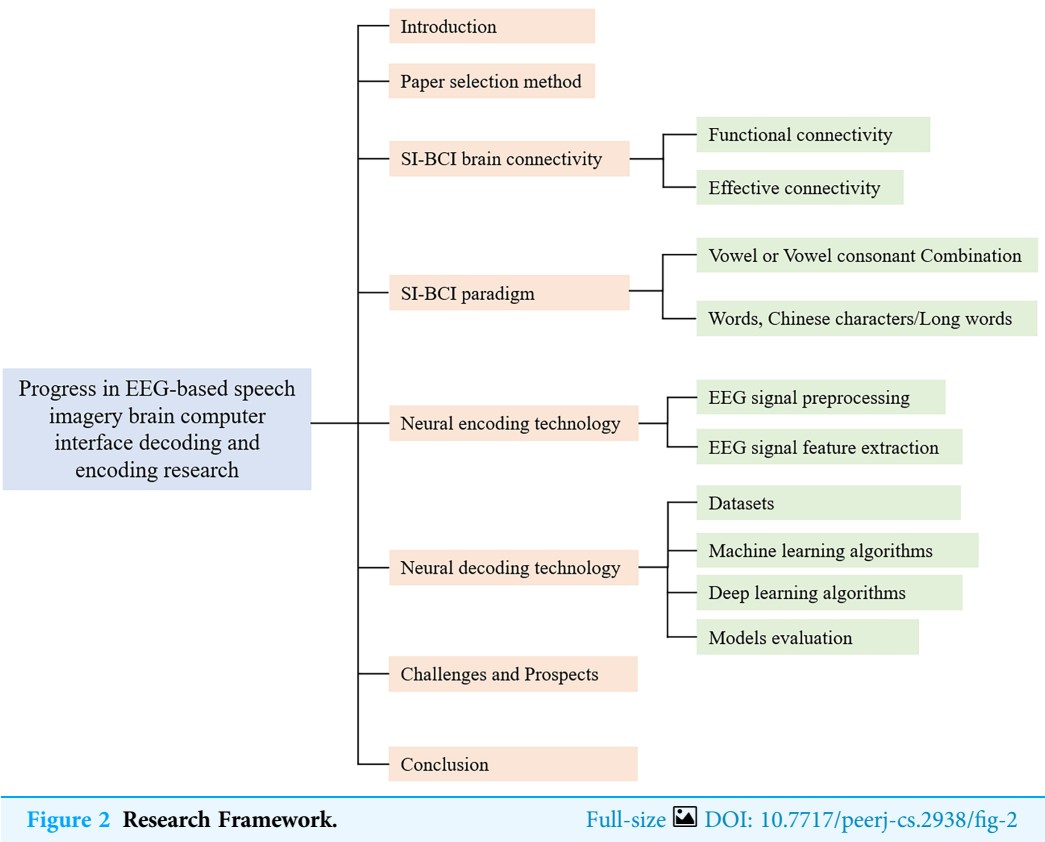

**Figure 2  Research Framework.**

analyses (*Page et al., 2021*). By using PRISMA to identify and locate relevant studies, the data collection process for this review was significantly streamlined.

The search was completed on March 8, 2025. We conducted queries using the following keywords on Web of Science, Scopus, IEEE Xplore, PubMed, ScienceDirect, and Google Scholar to gather relevant literature: ("brain-computer interface" AND ("speech imagery" OR "covert imagery" OR "inner speech" OR "imagined speech") AND ("deep learning" OR "machine learning" OR "DL" OR "ML") AND ("electroencephalogram" OR "EEG")). These keywords were sufficient to retrieve all pertinent studies.

Studies were excluded based on the following criteria:

After retrieving the search results, we excluded duplicate articles across databases and then applied the following criteria to filter out unsuitable studies from the remaining articles. Subsequently, we thoroughly read the full texts of the remaining studies.

(1) EEG-only studies: Studies using multimodal datasets were excluded.

(2) Language: Articles not written in English were excluded.

(3) Task type: Neural encoding and decoding tasks not exclusively based on EEG speech imagery were excluded.

(4) Time frame: Studies published before 2015 were excluded.

Ultimately, 125 highly relevant articles were selected for this literature review analysis. The distribution of these articles across brain connectivity, paradigms, neural encoding,

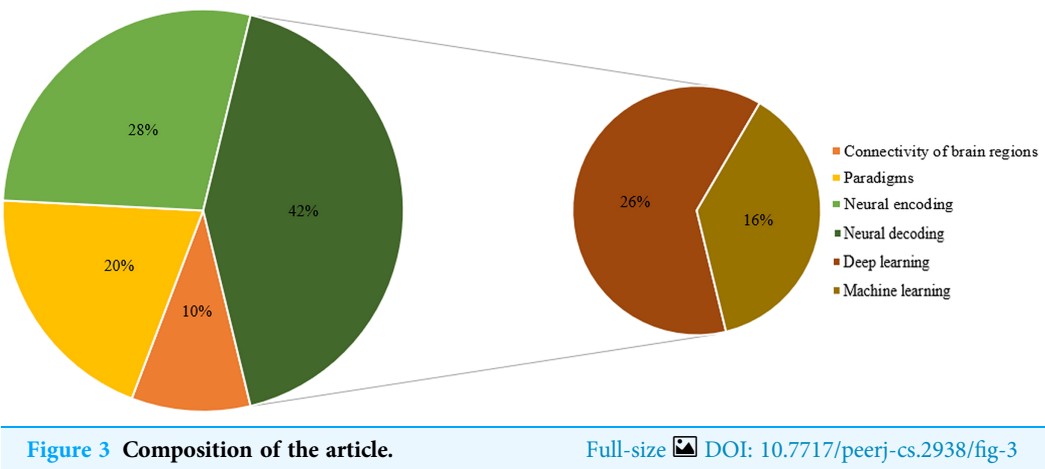

**Figure 3 Composition of the article.**

and neural decoding algorithms (machine learning and deep learning) is illustrated in
Fig. 3.

## CONNECTIVITY OF BRAIN REGIONS IN SI-BCI

Speech imagery, as a mental activity, involves the coordinated activity of multiple brain
regions. These regions are closely associated with language production and
comprehension, particularly Broca's area and Wernicke's area (*Jäncke, Liem & Merillat,
2021*; *Ono et al., 2022*). Brain region connectivity describes the information exchange and
interactions between these regions, playing a key role in both neural encoding and
decoding processes. In neural encoding, connectivity reflects the transmission and
collaboration of information between brain regions, supporting the encoding and
integration of complex information. In neural decoding, the use of connectivity features
can improve decoding performance and provide deeper insights into brain function
mechanisms. In the context of SI-BCI, brain region connectivity is primarily classified into
functional connectivity and effective connectivity.

Research on brain functional connectivity in speech imagery tasks is continuously
advancing to reveal the collaborative workings of different brain regions during specific
tasks or resting states. *Sandhya et al. (2015)* observed bilateral interactions between the
frontal and temporal regions of the brain during speech imagery by calculating the
coherence of EEG signals and applying a multivariate autoregressive (MVAR) model.
Notably, high inter-electrode coherence was observed in the left frontal lobe during
language production and in the left temporal lobe during speech imagery, highlighting the
proximity of these regions to Broca's area and Wernicke's area. This provides a
multidimensional perspective on understanding brain functional connectivity in speech
imagery, laying the foundation for subsequent research. With continued research,
*Chengaiyan & Anandhan (2015)*, *Chengaiyan et al. (2018)* further analyzed correlation
parameters of EEG signals and found that a single parameter is insufficient for
comprehensively understanding brain region connectivity. To address this, they calculated
connectivity parameters such as EEG coherence, partially directed coherence, and directed

transfer function, examining concurrent and directional connectivity patterns across different brain regions during speech production and speech imagery. Their findings indicated active engagement of the left frontal lobe during speech production and the left temporal lobe during silent word imagery, aligning with the roles of Broca's and Wernicke's areas and corroborating the results of *Sandhya et al. (2015)*. Additionally, phase locking value (PLV) calculations revealed the strongest phase synchronization between the left frontal and left temporal lobes in the alpha and theta bands, suggesting a close collaboration between these areas during speech/speech imagery. *Bisla & Anand (2023)* conducted a connectivity analysis of speech imagery using the Kara One dataset, identifying key neurophysiological dynamics in this paradigm. Notably, this was the first detailed exploration of neural connectivity related to speech imagery, highlighting distinct connectivity patterns through phase-based metrics and simplifying future data processing approaches.

To improve the accuracy of speech imagery recognition, *Chengaiyan & Anandan (2022)* innovatively combined brain connectivity metrics with machine learning methods, applying functional and effective connectivity analysis to a vowel recognition task, achieving over 80% recognition accuracy. This further demonstrates the value of brain connectivity analysis in understanding the mechanisms of speech cognition. However, traditional machine learning has limitations in studying brain region connectivity in speech imagery BCI research. It relies on manually designed feature extraction, which is time-consuming and susceptible to subjective influence. Traditional models also struggle to capture complex nonlinear relationships in data, especially when simulating intricate interactions among brain regions. Additionally, these models have limited capabilities with large-scale datasets and real-time applications, with weaker generalization. In contrast, deep learning addresses these limitations. It can automatically learn and extract features from raw data, reducing the need for manual feature engineering. Deep learning excels at capturing nonlinear patterns in data, enhancing prediction accuracy and providing strong generalization capabilities. *Park, Yeom & Sim (2021)* were the first to introduce deep learning into the field of brain connectivity, using mutual information as a measure of brain connectivity and applying convolutional neural networks (CNNs) for user state recognition tasks based on brain connectivity. By enabling CNNs to automatically learn connectivity features, they achieved adaptive control in BCI systems, offering a novel approach to developing adaptive BCI systems.

Effective connectivity is used to analyze the direct impact or causal relationships between activities in different brain regions, helping researchers understand the direction and dynamic processes of information flow in the brain (*Belaoucha & Papadopoulo, 2020*). Research primarily focuses on the design of brain connectivity estimators. *Sandhya et al. (2015)*, using brain connectivity estimators, analyzed causal correlations and found that the frontal and temporal regions of the left hemisphere were more active during specific speech imagery tasks. This represents significant progress in understanding neural interactions during thought and expression processes. *Panachakel & Ramakrishnan (2021)* used the mean phase coherence (MPC) as an indicator of cortical region phase synchrony and found that β-band MPC values differed significantly between nasal and bilabial

consonant imagery. This suggests that different speech imagery types lead to differences in effective connectivity between cortical regions. *Chengaiyan, Retnapandian & Anandan (2020)*, *Ahn et al. (2022)* applied deep learning to analyze effective connectivity in various brain regions during speech production and speech imagery, introducing an attention-based dual-modal fusion mechanism, which provides an innovative EEG signal analysis method for the fields of BCI and cognitive computing.

The aforementioned studies demonstrate that research on the neural mechanisms of speech imagery tasks focuses on the analysis of functional and effective connectivity between brain regions. This is primarily achieved through coherence, phase synchrony, and causal connectivity parameters derived from EEG signals, which reveal patterns of neural collaboration. In terms of technical methodologies, there is a clear shift from traditional machine learning approaches to deep learning techniques. These methodological advancements provide innovative frameworks for BCIs and cognitive computing, enabling a comprehensive understanding of the neural mechanisms underlying speech imagery. This progress lays a solid foundation for research on the SI-BCI paradigm.

## SI-BCI PARADIGM

The SI-BCI paradigms refer to the experimental design or task paradigm used in SI-BCI systems to elicit specific speech imagery-related brain signal patterns. It determines how users interact with the SI-BCI system and the types of signals generated by the brain. In SI-BCI systems, the BCI paradigms, neural encoding, and neural decoding are critical research elements. It is important to note that without BCI paradigms, the corresponding neural encoding cannot be achieved in an SI-BCI system; without appropriate BCI paradigms and neural encoding, high-performance neural decoding becomes challenging. Similarly, without effective neural decoding, the applicability of the previously designed BCI paradigms cannot be validated (*Tai et al., 2024*). Specifically, neural encoding refers to the process by which a user's different intentions are "written" or encoded into central nervous system signals under the SI-BCI paradigms. Neural decoding, on the other hand, is the process of extracting user intentions from neural signals in the SI-BCI system. It relies on the principles of neural encoding and identifies user intentions by analyzing and processing features within the neural signals. The SI-BCI paradigms not only influence neural encoding but also directly impact the performance of neural decoding. Figure 4 illustrates the relationships between brain region connectivity, the SI-BCI paradigms, neural encoding, and neural decoding in an SI-BCI system.

In an SI-BCI system, SI-BCI paradigms based on the type of imagined material can be divided into two experimental task paradigms:

1. Vowel or vowel-consonant combination: This is a basic binary classification task, focusing primarily on simple vowel imagery or combining consonants with vowels, such as through the consonant-vowel-consonant (CVC) structure in speech tasks. These experiments help explore the brain's neural encoding of fundamental speech units.

2. Words, Chinese characters/long words: Compared to vowels, words or longer speech segments like Chinese characters contain richer information and engage more complex

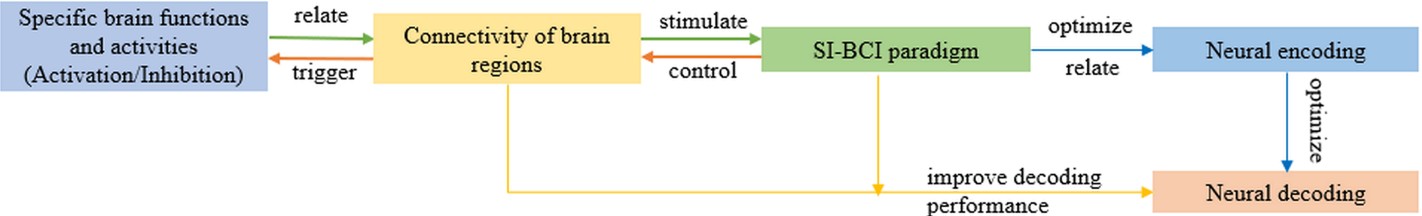

**Figure 4 The relationship between brain connectivity, SI-BCI paradigm, neural encoding, and neural decoding.**

neural networks. This paradigm also mainly uses binary classification, but with greater information complexity, making it suitable for more in-depth research on speech processing.

Figure 5 provides a detailed illustration of these two types of speech imagery BCI experimental paradigms and their characteristics, offering multi-level task options and research directions for the study of speech imagery BCI systems.

## Vowel/vowel consonant combination

In speech imagery-based BCI research, vowels are frequently used as core materials due to their simplicity in articulation, stability in acoustic characteristics, and ease of perception and measurement. Researchers have extensively explored the potential of EEG signals in decoding vowel information through diverse methods, progressively advancing BCI technology. *DaSalla et al.*'s *(2009)* pioneering work laid a solid foundation for vowel imagery recognition. By recording EEG signals from healthy subjects imagining the English vowels /a/ and /u/, they applied common spatial pattern (CSP) to design spatial filters and used nonlinear SVM for classification, achieving an accuracy rate of 68% to 78%. This study effectively addressed the challenge of extracting features related to vowel imagery from EEG signals, providing crucial methodological guidance for subsequent research. Building on this, *Min et al. (2016)* employed the Extreme Learning Machine (ELM) algorithm combined with sparse regression feature selection techniques to classify single-trial EEG signals from five subjects imagining vowels. They found that ELM and its variants (ELM-R, ELM-L) outperformed SVM and linear discriminant analysis (LDA) in the gamma frequency band (30–70 Hz), achieving a maximum accuracy of 99%. This work offered new insights into optimizing classification algorithms for silent speech BCIs. *Chengaiyan & Anandan (2022)* further refined vowel classification in speech imagery using EEG signal decomposition techniques and advanced machine learning methods such as multi-class SVM (MSVM) and RF. Their study demonstrated the reliability of brain connectivity estimators and machine learning techniques in vowel recognition, providing new empirical support for the development of speech imagery recognition technology. This series of studies not only progressively improved vowel imagery classification accuracy but also deepened our understanding of EEG signal processing and vowel imagery recognition mechanisms.

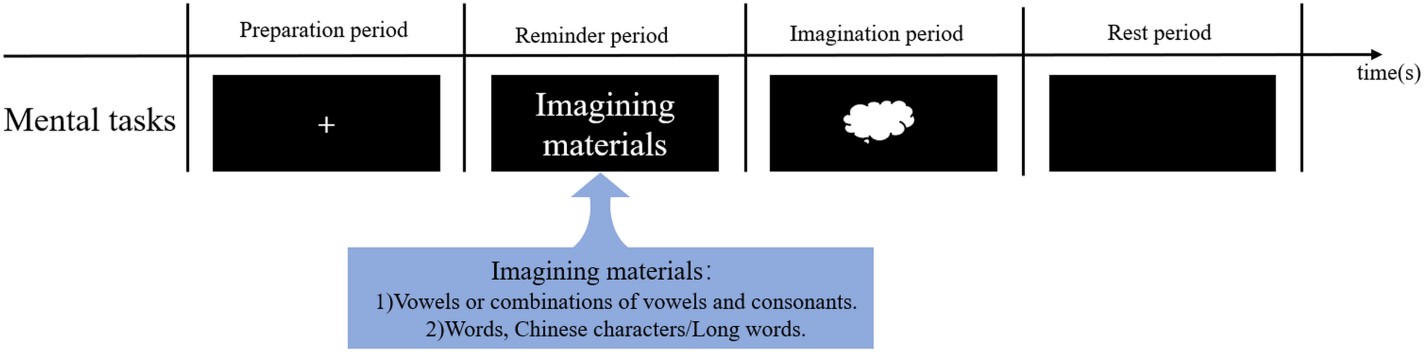

**Figure 5  The paradigm process of BCI in speech imagery.**

As research on vowel imagery has deepened, researchers have increasingly recognized the need to incorporate consonant processing into BCI systems to more comprehensively simulate natural speech communication. Given that actual speech communication involves not only vowels but also consonants, particularly the prevalence of CVC structured words in the language, researchers have begun exploring how to decode combinations of consonants and vowels in BCI systems. This direction represents a significant step forward in the development of speech imagery-based BCI systems.

The work by *Sandhya et al. (2018)* marks an early exploration in this field. They investigated relative power changes during multiple speech imagery tasks, with a particular focus on frequency bands in EEG signals. By analyzing EEG signals from healthy subjects imagining and articulating CVC words, they found that the theta band exhibited higher relative power during imagined speech, while the alpha band dominated during actual speech. This suggests that different speech imagery tasks may activate distinct frequency bands in the brain. Building on the work of *Sandhya et al. (2018)*, *Chengaiyan et al. (2018)* further explored the power and phase synchronization effects in EEG signals. Using the PLV to measure phase synchrony between brain regions during speech imagery, they discovered that the frontal and temporal electrodes in the left hemisphere showed the highest phase locking in the alpha and theta bands. This work highlights the critical role of functional connectivity between specific brain regions in decoding tasks during speech imagery. *Sandhya et al. (2015)* focused on analyzing the neural correlations of brain regions during the speech imagery of CVC words. Using brain connectivity estimators such as EEG coherence, partial directed coherence (PDC), and directed transfer function (DTF), they found that the frontal and temporal regions of the left hemisphere were more active during both imagined and actual articulation. This study provides new insights into the neural interactions in the brain during speech imagery and offers neurological support for decoding CVC words. *Chengaiyan, Retnapandian & Anandan (2020)* further advanced this direction by integrating deep learning techniques, such as recurrent neural networks (RNN) and deep belief networks (DBN), with brain connectivity estimators to focus on vowel recognition from EEG signals. In their experiments, DBN demonstrated higher

classification accuracy compared to RNN, providing strong empirical support for the use of deep learning techniques in speech imagery decoding.

From the initial CSP and SVM-based classification methods to advanced techniques involving deep learning networks (such as CNN, RNN, DBN, and capsule networks (CapsNet)) and brain connectivity analysis, researchers have continuously deepened their understanding of vowel and vowel-consonant combination information in EEG signals. The development of these technologies has not only complemented each other methodologically but has also made significant progress in improving classification accuracy and the practicality of BCI systems.

## Chinese characters/words/long words

In studies using Chinese characters as materials for pronunciation imagery, directional characters such as "左(left)", "右(right)" have become the primary focus. As research has progressed, the emphasis has shifted from simple characters to those with tonal and rhythmic features, and from single speech imagery tasks to those combining auditory and visual conditions. In research on words and word pairs, although the materials differ, the mental tasks are similar. Notably, the use of local ear EEG as an alternative to global EEG has emerged as a novel approach.

*Wang et al. (2016)* made pioneering contributions in this field by introducing and applying the novel concept of "speech imagery" to BCI systems. They conducted a study to explore the impact of silent reading on mental tasks within a BCI system. They integrated speech imagery (*i.e.*, silently reading Chinese characters) into mental tasks and found that adding speech imagery significantly improved task accuracy, with the average accuracy increasing from 76.3% to 82.3%. They also evaluated the time stability of EEG signals using Cronbach's alpha, revealing that the tasks incorporating speech imagery exhibited higher signal stability, providing a more reliable signal source for BCI systems. Furthermore, the classification of Mandarin tones and word pairs has also received widespread attention. *Li & Chen (2020)* conducted in-depth studies on Mandarin tone classification, analyzing factors that affect the classification accuracy of EEG signals. This work laid the foundation for the application of BCI technology in Mandarin speech synthesis systems and contributed to the theoretical development of speech-imagery-based BCI systems. *Borirakarawin & Punsawad (2023)* explored hybrid BCI systems that combine auditory stimuli with speech imagery, while *Zhang, Li & Chen (2020)* further explored tone classification under audiovisual conditions, and their experimental results showed that the introduction of audiovisual conditions significantly improved classification accuracy, demonstrating that multimodal inputs can effectively enhance BCI system performance and provide new experimental evidence for Mandarin tone-based BCI applications.

In research on words and word pairs, innovative algorithms and techniques have emerged continuously. *Mahapatra & Bhuyan (2022)* achieved multi-class imagined speech (vowels and words) classification based on EEG signals by constructing a deep model that integrates temporal convolutional networks (TCN) with CNN, combined with discrete wavelet transform (DWT) preprocessing. They achieved an overall accuracy of 96.49% on data from 15 participants, providing an efficient decoding solution for non-invasive silent

brain-computer interfaces. *Pan et al. (2023)* employed a light gradient boosting machine (LightGBM) for feature classification in Chinese character speech imagery BCI systems, and the results showed that LightGBM was more accurate and applicable than traditional classifiers. *Tsukahara et al. (2019)* explored EEG frequency components related to speech imagery and found significant ERS in the left hemisphere's alpha band under speech imagery conditions, emphasizing the importance of selecting appropriate electrode positions for EEG frequency component recognition. *Kaongoen, Choi & Jo (2021)* were the first to explore BCI systems based on ear EEG (ear-EEG), providing initial empirical support for the application of ear-EEG in the BCI field.

These studies not only explored various EEG signal processing and feature extraction methods but also experimented with multiple classifiers to improve the decoding accuracy of EEG signals. Furthermore, they considered the complexity of speech imagery, such as semantics, rhythm, and brain activity under different stimulus conditions. These efforts are particularly significant in optimizing BCI system design, improving signal decoding accuracy, and achieving effective classification of Mandarin tones and word pairs.

## SI-BCI NEURAL ENCODING

SI-BCI neural encoding refers to the process of translating distinct user intentions into central nervous system signals within the speech imagery paradigm, characterized by identifiable brain signal features (*Xu et al., 2021*). EEG technology is employed to detect brain signals that encode user intentions, which are subsequently decoded by SI-BCI neural decoding algorithms to identify these intentions. Brain signals acquired through EEG technology exhibit multiple features in the time domain, frequency domain, and spatial domain. Therefore, SI-BCI neural encoding can utilize these three types of features for encoding, such as mean, variance, power spectral density (PSD), wavelet transform (WT), short-time Fourier transform (STFT), DWT, and CSP. Figure 6 illustrates the process of SI-BCI neural encoding.

The SI-BCI neural encoding process involves three main steps: EEG signal acquisition, preprocessing, and feature extraction. Depending on the research context, EEG signal acquisition may involve electrode caps with configurations ranging from eight channels to more than 64 channels. Common acquisition devices include Neuroelectrics (NE), SynAmps 2, TMSi SAGA, and EMOTIV, among others. As the signal acquisition process predominantly relies on hardware, the quality of the acquired signals can be significantly influenced by the experimenter's proficiency. Thus, we present only common electrode cap configurations and device types to offer essential references for researchers.

### Signal preprocessing

EEG signal preprocessing involves filtering out irrelevant or interfering signals from the raw EEG data acquired by the device and selecting the signal channels most relevant to speech imagery from specific brain regions. Due to the low signal-to-noise ratio (SNR) of EEG signals, artifacts often occur during acquisition. Therefore, current research focuses on using techniques such as filtering, wavelet transforms, adaptive filters, and statistical methods to remove noise and artifacts, thereby improving signal quality. *Jahangiri &*

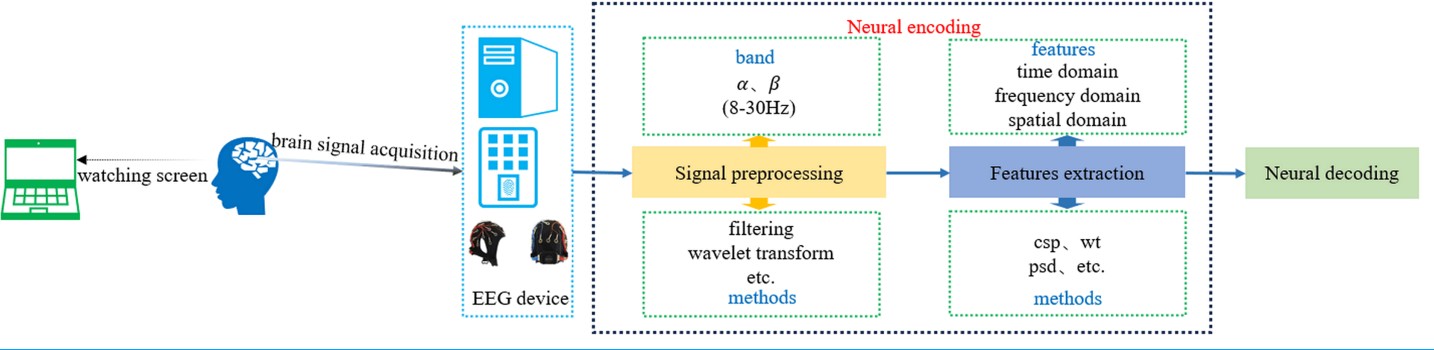

**Figure 6 SI-BCI neural encoding process.**

*Sepulveda (2017)* found that the alpha band (8–13 Hz) and beta band (14–30 Hz) yield higher classification performance in speech imagery classification tasks. Consequently, during time-domain and time-frequency analysis, signals are typically filtered within the 8–30 Hz range, with different bandpass filters selected based on specific requirements. *Moattari, Parnianpour & Moradi (2017)* proposed using independent component analysis (ICA) based on higher-order non-Gaussianity for the source separation stage of preprocessing. *Nitta et al. (2023)* extracted linguistic representations of Japanese vowels using principal component analysis (PCA) and structural modeling (SM), and demonstrated the feasibility of extracting linguistic features from EEG signals by classifying them with a CNN. *Sree, Kavitha & Divya (2023)* evaluated various preprocessing methods for EEG signals based on speech imagery. Through a comprehensive comparison of different preprocessing techniques, they identified the optimal approach based on mean squared error (MSE) and peak signal-to-noise ratio (PSNR).

## Feature extraction

The feature extraction stage is the core of SI-BCI neural encoding. This process involves extracting useful information from the acquired EEG signals, which is then used in subsequent neural decoding to analyze brain states. Feature extraction algorithms can be categorized into spatial domain methods, time domain methods, frequency domain methods, time-frequency domain methods, and nonlinear feature extraction methods.

In the spatial domain, the main research trend has shifted from traditional CSP and their variants to methods such as Mel-frequency cepstral coefficients (MFCCs). Overall, new methods in the spatial domain are continuously emerging. *Wang et al. (2021)* proposed a feature extraction method combining causal networks and CSP, achieving higher classification accuracy using a binary quantum particle swarm optimization-based extreme learning machine. *Huang et al. (2022)* introduced a novel algorithm called sub-time window filter bank common spatial pattern (STWFBCSP), aimed at improving sequence encoding-based active BCIs. By subdividing time windows and applying multi-frequency filtering, STWFBCSP can extract more refined features, thereby enhancing classification accuracy. The final average classification accuracy of STWFBCSP

reached 84.87%, compared to 70.49% for the traditional CSP algorithm, demonstrating its superior performance and highlighting the significant potential of time window subdivision and multi-frequency analysis in improving BCI performance. *Alizadeh & Omranpour (2023)* proposed a novel multi-class CSP feature extraction method, feeding the extracted features into an ensemble learning classifier (including logistic regression, K-nearest neighbors, decision trees, SVM, and Gaussian naive Bayes). After multiple rounds of cross-validation, the effectiveness of the proposed algorithm was confirmed. This combined validation approach effectively inspires algorithmic fusion and innovation. Table 1 illustrates CSP combined with other feature classification algorithms.

*Cooney, Folli & Coyle (2018)* evaluated the impact of three feature sets—linear, nonlinear, and MFCCs—on classification performance. The study found that MFCC features performed best in both decision tree and SVM classifiers, with SVM achieving an average accuracy of 20.80% on MFCC features, significantly outperforming linear (15.91%) and nonlinear (14.67%) features. This validates the effectiveness of MFCCs in capturing acoustic differences in speech. *Martin et al. (2016)* introduced the dynamic time warping (DTW) algorithm to address temporal irregularities in speech production, improving the classification performance of SVM. Notably, this marked the first successful classification of individual words during speech imagery tasks. *Wu & Chen (2020)* proposed a method for extracting temporal envelope features from EEG signals during feature extraction and directly classified these features using SVM. Although the final performance was somewhat lacking compared to other methods, this work explored the relationship between temporal envelope features and speech. Table 2 illustrates other feature extraction algorithms in the spatial domain.

Time-domain methods generally involve selecting features such as the mean, variance, kurtosis, and skewness of signals from each channel. Research in this area primarily focuses on extracting features using wavelet transforms and wavelet decomposition for signals of different frequencies. *Idrees & Farooq (2016a, 2016b)* proposed a dual-pronged strategy: classifying using simple features in the time domain while extracting EEG signal features from the beta, delta, and theta rhythms using wavelet decomposition. This approach demonstrated the effectiveness of time-domain and frequency-domain features in vowel imagery classification, further expanding the potential of EEG signal processing techniques in speech imagery tasks. *Saji et al. (2020)* advanced this research by classifying vowel signals using wavelet transforms and multiple classifiers, extracting features such as mean, standard deviation, and band power. This study not only confirmed the effectiveness of EEG signal processing techniques in improving vowel imagery classification accuracy but also provided new empirical support for speech imagery-based BCI technology.

Frequency-domain methods have been more deeply explored in areas such as PSD, DWT, and MFCCs. *Cooney, Folli & Coyle (2018)* analyzed the EEG signals of imagined speech from 14 participants in the Kara One dataset, evaluating the impact of three feature sets—linear, nonlinear, and MFCCs—on classification performance. The study found that MFCC features performed best in both decision tree and SVM classifiers, with SVM achieving an average accuracy of 20.80% on MFCC features, significantly outperforming

**Table 1 CSP combined with other feature classification algorithms.**

| A | M | P | D | S | DE | E |
|---|---|---|---|---|---|---|
| *DaSalla et al. (2009)* | CSP SVM | /a/, /u/ | Private data | 3 (2 m and 1 f/age from 26–29) | BioSemi ActiveTwo | 68–78 |
| *Agarwal et al. (2020)* | CSP SVM | /a/, /u/, /rest/ | *DaSalla et al. (2009)* | 3 (2 m and 1 f/age from 26–29) | BioSemi ActiveTwo | 87.56 |
| *Rostami & Moradi (2015)* | EMBCSP SVM | /a/, /u:/ | Private data | 5 (3 m and 2 f/age from 23–33) | N/A | Average: 89.42 |
| *Wang et al. (2016)* | CSP SVM | Chinese character "左 (Left), 壹(One)" | Private data | 10 (7 m and 3 f/age from 22–28) | SynAmps 2 | Average: 82.3 |
| *Wang et al. (2019)* | CSP, CCF, PLV SVM | Chinese character "移 (Move)" | Private data | 10 (8 m and 2 f/age from 22–28) | SynAmps 2 | Average: 74.3 |
| *Zhang, Li & Chen (2020)* | CSP SVM | /ba/ | Private data | 14 (6 m and 8 f/age from 19–22) | N/A | Under combined stimuli: 80.1 |
| *Wang et al. (2021)* | CSP SVM | Chinese character "移 (Move)" | Private data | 10 (8 m and 2 f/age from 22–28) | SynAmps 2 | Average: 73.9 |
| *Zhao, Liu & Gao (2021)* | CSP, DWT SVM, ELM | Chinese initial consonant | Private data | 8 (6 m and 2 f/ average age 23.67) | N/A | Highest: 73.04 |
| *Wang et al. (2022)* | CSP SVM | Chinese character "壹 (One)" | Private data | 12 (9 m and 3 f/age from 22 to 26) | Neusen W2 | Average: 68.94 |

**Note:**

A, authors; M, methods; P, pronunciation materials; D, datasets; S, subjects (number); DE, device; E, evaluation indicators (accuracy: %); m, males; f, females; CSP, common spacial pattern; SVM, support vector machine; EMBCSP, evidential multi-band common spacial pattern; CCF, cross correlation function; PLV, phase locking value; DWT, discrete wavelet transform; ELM, Extreme Learning Machine; SI, speech imagery; MI, motor imagery.

**Table 2 Other feature extraction algorithms in spatial domain.**

| A | M | P | D | S | DE | E |
|---|---|---|---|---|---|---|
| *Wang et al. (2021)* | CN, CSP BQPSO ELM | Chinese characters "壹(one)" | Private data | 10 (7 m and 3 f/age from 22–28) | SynAmps 2 | Average: 85.4 |
| *Huang et al. (2022)* | STWFBCSP | Chinese characters "右(right)" | Private data | 12 (8 m and 4 f/age from 22–26) | N/A | Average: 84.87 |
| *Alizadeh & Omranpour (2023)* | EM-CSP | Words, English phonemes | 1. Kara One (*Zhao & Rudzicz, 2015*) 2. Private data | 12 (8 m and 4 f/age mean 27.4) | SynAmps RT | Highest: 98.47 |
| *Panachakel & Ramakrishnan (2022)* | CSP LSTM | "in, cooperate" | ASU (*Nguyen, Karavas & Artemiadis, 2017*) | 15 (11 m and 4 f/age from 22–32) | BrainProducts ActiCHamp amplifier | Highest: 85.2 |

**Note:**

A, authors; M, methods; P, pronunciation materials; D, datasets; S, subjects (number); DE, device; E, evaluation indicators (accuracy: %); m, males; f, females; CN, causal network; CSP, common spatial pattern; BQPSO, binary quantum particle swarm optimization; ELM, Extreme Learning Machine; STWFBCSP, sub-time window filter bank common spatial pattern; EM-CSP, efficient-multiclass CSP; LSTM, long short-term memory; SI, speech imagery; MI, motor imagery.

linear (15.91%) and nonlinear (14.67%) features. This research demonstrated that MFCC-based feature extraction can significantly improve the accuracy of imagined speech decoding, providing important methodological support for the development of SI-BCIs. Time-frequency domain methods combine information from both the time and frequency domains, commonly using techniques such as STFT and WT. *Pan et al. (2022)* investigated the impact of imagined syllable rhythms on EEG amplitude spectra, offering new insights for the development of speech imagery-based BCIs. *Kamble, Ghare & Kumar (2022)* preprocessed EEG signals through bandpass filtering and downsampling in the time domain, combined with time-frequency conversion using the smoothed pseudo-Wigner-Ville distribution (SPWVD). This mapped 1D signals into 2D time-frequency images, enabling optimized CNNs to automatically extract multi-scale time-frequency features. This approach achieved efficient speech imagery classification with a maximum accuracy of 94.82%, validating the advantages of SPWVD in time-frequency resolution and cross-term suppression. It provided a time-frequency analysis-based method for SI-BCI systems.

Nonlinear feature extraction methods refer to the use of deep learning to automatically extract features for subsequent classification tasks. Unlike traditional machine learning, deep learning-based feature extraction methods are often not separable from feature classification methods. This is primarily because, in contrast to deep learning, traditional machine learning requires manual intervention during feature extraction to transform raw data into meaningful features. The core focus of deep learning research lies in developing more efficient and convenient algorithms for feature extraction while maximizing final classification performance. *Wang et al. (2022)* proposed a parallel CNN model based on a multi-band brain network, utilizing correlation coefficients and phase-locking values to describe inter-channel synchrony and correlation. This approach effectively captures EEG signal synchronization and correlation features, leading to improved classification accuracy. *Ramirez-Quintana et al. (2023)* introduced a novel deep capsule neural network, CapsVI, designed to recognize vowel imagery patterns from EEG signals. Their model achieved an average accuracy of 93.32%, setting a new benchmark for English vowel recognition. *Retnapandian & Anandan (2023)* focused on identifying phonemes corresponding to vowels from EEG signals. They trained a RNN using multi-trial data and compared the classification performance between single-trial and multi-trial datasets. By incorporating refined feature extraction techniques such as wavelet decomposition and multifractal analysis, their study further enhanced the classification capabilities of the model, laying the foundation for the development of efficient and precise speech imagery-based BCI systems. *Macías-Macías et al. (2020)* developed sCNN and sCapsNet models based on spectro-temporal receptive fields (STRFs). Notably, sCapsNet demonstrated superior performance in vowel classification, highlighting the significant potential of deep learning in vowel imagery classification.

In addition to the commonly used feature extraction algorithms mentioned above, several novel methods have also been applied to SI-BCI feature extraction. *Sikdar et al. (2018)*, *Sikdar, Roy & Mahadevappa (2018)* explored the roles of multifractal and chaotic parameters in different imagery tasks for International Phonetic Alphabet (IPA) vowel

recognition, providing new analytical methods for identifying brain regions activated during different vowel imagery processes. *Martin et al. (2016)* utilized high-gamma band (70–150 Hz) temporal features and a SVM model, introducing nonlinear temporal alignment *via* an SVM kernel, achieving 88% classification accuracy in a binary classification framework. *Nguyen, Karavas & Artemiadis (2017)* proposed a novel approach based on covariance matrix descriptors, which reside on the Riemannian manifold, and employed a relevance vector machine (RVM) for classification. This method was tested on EEG signals from multiple subjects and performed well across various categories, including imagined vowels, short words, and long words, with classification accuracy significantly exceeding chance levels in all cases. *Kalaganis et al. (2023)* leveraged approximate joint diagonalization (AJD) for covariance estimation, further advancing research on decoding human speech imagery from EEG signals.

In summary, during SI-BCI neural decoding, optimizing the performance of traditional algorithms (*e.g.*, CSP, RF) while further exploring deep learning-based approaches (*e.g.*, CNN, RNN) has become a key trend in EEG signal preprocessing and feature extraction. Meanwhile, unconventional algorithms (*e.g.*, covariance matrix-based methods) have also garnered increasing attention.

## SI-BCI NEURAL DECODING

SI-BCI neural decoding refers to the process of identifying a user's intent by employing efficient decoding algorithms on neural-encoded brain signals, which are then translated into syllables, words, Chinese characters, or other linguistic units. These decoded outputs are subsequently presented through visual or auditory modalities (*Saha & Fels, 2019*). Figure 7 illustrates the SI-BCI neural decoding process.

Currently, based on different decoding algorithms, the current SI-BCI decoding methods can be divided into two categories: machine learning and deep learning. In machine learning algorithms, the feature extraction and feature classification steps are separate, so innovative combinations of different types of feature extraction algorithms and feature classification algorithms have led to numerous high-performance decoding algorithms. It is worth noting that research using CSP (*Cai et al., 2024*) (feature extraction) combined with SVM (*Rezvani et al., 2024*) (feature classification) as baseline methods has always been highly praised. Additionally, methods such as wavelet decomposition (*Ji et al., 2024*), covariance matrices (*Luo et al., 2024*), relevance vector machines (*Khatti & Grover, 2024*), and extreme learning machines (*Wei et al., 2024*) have also demonstrated unique performance values. Deep learning algorithms, on the other hand, use one or more neural network models to complete both feature extraction and feature classification. Among these, convolutional neural networks (*Liang et al., 2023*) have achieved excellent performance in SI-BCI neural decoding. In addition, capsule networks (*Aboussaleh et al., 2024*) and gated mechanisms (*Li et al., 2024*) have also shown good potential.

However, in the study of SI-BCI decoding methods, the performance of decoding algorithms depends not only on the characteristics of the algorithms themselves but also closely on the quality and diversity of the data. Both machine learning and deep learning approaches require large amounts of high-quality data for model training and validation.
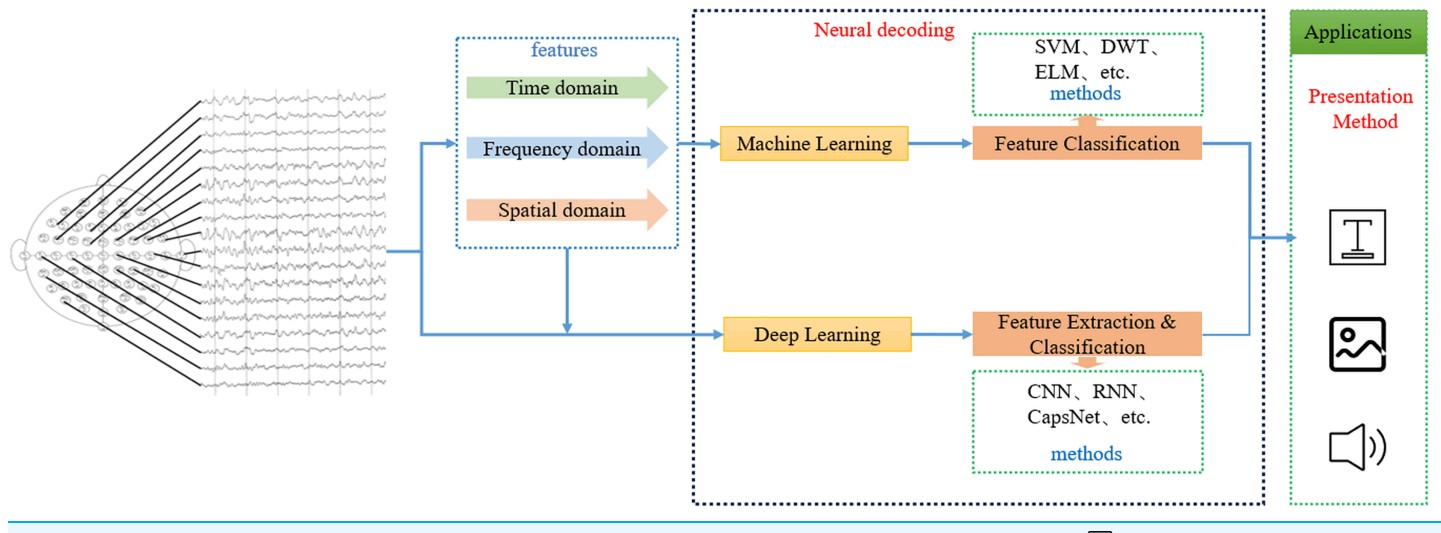

**Figure 7** SI-BCI neural decoding process.

Therefore, before systematically reviewing neural encoding, understanding the current usage of existing datasets is of significant importance for further advancing SI-BCI technology.

## Datasets

We have compiled statistics on existing public and private SI-BCI datasets, and Fig. 8 visually presents their usage in percentage. Current research predominantly relies on private datasets from individual laboratories, with public dataset usage being relatively limited. This phenomenon is influenced by several factors, including the complexity of data collection, the need for data quality assurance, and limitations in participant numbers. Among these, the most frequently used public datasets are the Kara One, DaSalla, and ASU datasets. Thus, we provide an introduction to these three primary datasets, along with a recently released high-channel EEG dataset based on Chinese language stimuli.

**Kara One** (*Zhao & Rudzicz, 2015*): The Kara One dataset was collected and organized by the Department of Computer Science at the University of Toronto in collaboration with the Toronto Rehabilitation Institute. It integrates three modalities (EEG, facial tracking, and audio) during imagined and vocalized phonemes and single-word prompts, providing access to the brain's language and speech production centers. Each participant was seated in a chair in front of a computer monitor. A Microsoft Kinect (v.1.8; Microsoft Corp., Redmond, WA, USA) camera was placed beside the screen to record the participant's facial information and speech. Each trial consisted of four consecutive states:

1. A 5-s rest state, during which participants were instructed to relax and clear their minds.

2. A stimulus state, where a prompt text appeared on the screen, and its associated auditory utterance was played through computer speakers. This was followed by a 2-s

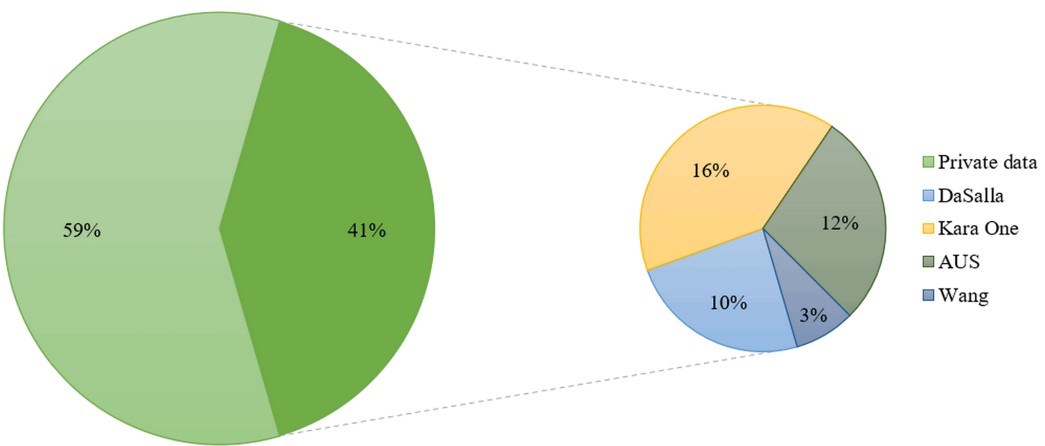

**Figure 8** Proportion of usage between public and private datasets.

period during which participants prepared their articulatory posture to begin vocalizing the prompt.

3. A 5-s imagined speech state, during which participants imagined speaking the prompt without actual movement.

4. A speaking state, where participants vocalized the prompt aloud. The Kinect sensor recorded audio and facial features during this phase.

**ASU** (*Nguyen, Karavas & Artemiadis, 2017*): The ASU dataset, provided by Arizona State University, is widely used for research and development in EEG-based BCI systems, particularly in identifying and decoding brain activity patterns related to imagined speech. A key feature of this dataset is that it includes EEG signals from multiple subjects performing specific imagined speech tasks. Participants were instructed to imagine hearing words or phrases rather than vocalizing them. The dataset comprises multiple EEG channels to capture brain activity. The task design incorporates various conditions, such as imagining different words or phrases, which aids researchers in exploring the patterns of brain activity during the process of speech imagination.

**DaSalla** (*DaSalla et al., 2009*): The DaSalla dataset was compiled and contributed by DaSalla and their team. This dataset generates speech-related potentials (SRP) in EEG signals for imagined vowels /a/ and /u/. Compared to other dataset's imagined speech paradigms, SRP exhibits better variance. Each trial lasts approximately 1 min. It begins with a beep at 0 s to prepare the subject for the first step. The trial then proceeds through three stages:

1. A cross appears on the display for a random duration of either 2 or 3 s to prevent erroneous event-related potentials (ERPs) in subsequent steps.

2. Vowel imagination is performed in this step. It is represented by an open mouth outline, a rounded mouth, or a cross, indicating the /a/, /u/, or control state imagination tasks, respectively. This step lasts for 2 s. In this study, the control state is referred to as /no

vowel/ since it involves no vowel. Each subject performed 150 trials, with 50 trials per task. The presentation of /a/, /u/, and /no vowel/ imagination tasks were randomized to train subjects to develop speech imagination tasks solely through endogenous stimuli.

3. A blank image is displayed on the screen for 3 s to indicate a rest period and the end of the trial, allowing the subject to wait for the next trial.

**ChineseEEG** (*Mou et al., 2024*): The ChineseEEG dataset is a high-channel EEG dataset based on Chinese language stimuli, providing a rich resource for semantic alignment and neural decoding. It includes high-density EEG data and synchronized eye-tracking data from 10 participants during approximately 13 h of silent reading of Chinese texts. The data is derived from the Chinese versions of two well-known novels, The Little Prince and Dream of the Wolf King. Each participant was exposed to diverse Chinese language stimuli, which is crucial for studying the long-term neural dynamics of language processing in the brain. Additionally, the 128-channel high-density EEG data offers superior spatial resolution, enabling precise localization of brain regions involved in language processing. With a sampling rate of 1 kHz, the dataset effectively captures the dynamic changes in neural representations during reading.

## Machine learning based SI-BCI neural decoding

Most studies based on machine learning employ CSP for feature extraction and SVM for feature classification. The principle of SVM is to use mathematical optimization methods to find a hyperplane that maximizes the separation of two classes of data (*Rezvani et al., 2024*). The principle of CSP is to find a set of spatial filters that maximize the variance differences between different categories of EEG signals (*Cai et al., 2024*). This is the most common baseline approach. *DaSalla et al. (2009)*, using the method of CSP combined with SVM, successfully achieved a classification accuracy rate of 68% to 78% for imagined pronunciations of the English vowels /a/ and /u/, laying the foundation for this field. *Agarwal et al. (2020)* combined CSP, statistical features (standard deviation, root mean square (RMS), energy), and wavelet transforms and employed a Random Forest classifier for multi-class classification of imagined vowels /a/, /u/, and the rest state. They achieved a maximum accuracy of 89% on data from three subjects, demonstrating the effectiveness of integrating CSP with statistical features in silent speech decoding. *Rostami & Moradi (2015)* proposed combining CSP with multiple bandpass filters and evidence theory, achieving a classification accuracy superior to traditional CSP methods. Table 3 shows the use of CSP and SVM along with their derived algorithms.

In addition to the aforementioned methods, several other feature classification approaches have also garnered significant attention. *Idrees & Farooq (2016a)* designed a vowel classification method based on wavelet decomposition and statistical features, achieving significant results with three subjects. *Nguyen, Karavas & Artemiadis (2017)* proposed a method based on Riemannian manifold features and RVM, achieving high accuracy in binary classification, but the accuracy in three-class classification significantly decreased. To improve the accuracy of three-class classification, they proposed a new method combining spatial covariance with an adaptive weighting model. Experimental

**Table 3 Adopting CSP, SVM and their derived algorithms.**

| A | M | P | D | S | DE | E |
|---|---|---|---|---|---|---|
| Cooney, Folli & Coyle (2018) | MFCCs, SVM, | /iy/, /uw/, /piy/, /tiy/, /diy/, /m/, /n/ | Kara One (Zhao & Rudzicz, 2015) | 12 (8 m and 4 f/age mean 27.4) | SynAmps RT | 35 |
| | | "pat", "pot", "knew", "gnaw" | | | | |
| Martin et al. (2016) | TDC SVM +DTW | Words | Private data | 5 | g.USBamp | Average: 57.7 Highest: 88.3 |
| Wu & Chen (2020) | TE | /iy/, /uw/, /piy/, /tiy/, /diy/, /m/, /n/ | Kara One (Zhao & Rudzicz, 2015) | 12 (8 m and 4 f/age mean 27.4) | SynAmps RT | Normalized covariance: 0.57 |
| | SVM | "pat", "pot", "knew", "gnaw" | | | | |
| Chengaiyan & Anandan (2022) | MSVM RF | /a/, /e/, /i/, /o/, /u/ | Private data | 5 (5 m / age from 19–21) | Emotiv Epoc wireless EEG | Highest: 80 |

Note:
A, authors; M, methods; P, pronunciation materials; D, datasets; S, subjects (number); DE, device; E, evaluation indicators (accuracy: %); m, males; f, females; MFCCs, Mel-frequency cepstral coefficients; AC, autoregressive coefficient; SVM, support vector machines; TDC, time domain characteristics; DTW, dynamic time warping; TE, temporal envelope; MSVM, multi-class support vector machine; RF, random forest.

results showed significant improvement, and surprisingly, subjects from the BCI blind group also achieved good results. Table 4 summarizes other machine learning feature extraction and classification algorithms.

## Deep learning based SI-BCI neural decoding

There are numerous algorithms in deep learning, and within the field of SI-BCI, CNN, RNN, and capsule neural networks have garnered significant attention. CNNs have shown significant performance in image processing. They are deep, feedforward artificial neural networks. CNNs consist of an input layer, several hidden layers, and an output layer, with trainable weights and biases used to construct the neural framework. Each neuron receives data and then performs a dot product using a nonlinear function. The hidden layers include a series of convolutional layers and pooling layers, which multiply or perform dot product convolutions in other ways. Figure 9 is a typical CNN model generated using our visualization tool.

Research on CNNs tends to combine them with existing feature extraction methods, feeding the extracted features into the CNN to capture finer details. Saha & Fels (2019) proposed a novel hybrid deep neural network architecture for EEG speech imagery classification tasks. It is noteworthy that this was the first time CNNs were introduced into the field of speech imagery brain-computer interfaces, leading to groundbreaking advancements in subsequent research. Macías-Macías et al. (2020) first used CSP to extract signal features, then employed CNNs for feature classification. This combination effectively avoided the problem of CNNs losing significant features during the feature extraction process. Lee et al. (2021) used a CNN model to predict the word length corresponding to the input EEG signal and explored the impact of word length on speech imagery classification performance, concluding that training strategies based on limited word length could effectively improve overall classification accuracy. Park, Yeom & Sim

**Table 4 Summary of other machine learning feature extraction and feature classification algorithms.**

| A | M | P | D | S | DE | E |
|---|---|---|---|---|---|---|
| *Idrees & Farooq (2016a)* | TDC<br>LC | /a/, /u/, other | Private data | 3 (2 m and 1 f) | BioSemi ActiveTwo | Combination average: 85–100 |
| *Idrees & Farooq (2016b)* | WD<br>LC | /a/, /u/, "no" | Private data | 3 (2 m and 1 f) | BioSemi ActiveTwo | Combination average: 81.25–98.75 |
| *Moattari, Parnianpour & Moradi (2017)* | HON-ICA | /a/, /u:/ | *Rostami & Moradi (2015)* | 5 (3 m and 2 f/age from 23–30) | N/A | 66.67–93.33 |
| *Nguyen, Karavas & Artemiadis (2017)* | CM<br><br>RVM | "/a/,/i/,/u/"<br>"in, out, up"<br>"cooperate, independent" | Private data | 15 (11 m and 4 f/age from 22–32) | BrainProducts ActiCHamp amplifier | Highest: 95 (Binary classification), 70 (Three categories) |
| *Nguyen, Karavas & Artemiadis (2019)* | SCM<br><br>RVM | Long word: "concentrate"<br>Short word: "split" | Private data | 8 (6 m and 2 f/age from 22–32) | BrainProducts ActiCHamp amplifier | Average: 52.5 |
| *Kim, Lee & Lee (2020)* | ERP<br>RLDA | "Ah", Specific nouns | Private data | 2 (2 f/age from 22–27) | ActiCap EEG amplifier | Highest combination: 88.1 |
| *Wang et al. (2020)* | PSD, SampEn ELM | Chinese character "移(move)" | Private data | 12 (8 m and 4 f/age from 20–26) | SynAmps 2 | Average: 83 |
| *Pan et al. (2023)* | WPD<br>LightGBM | Chinese character "左(Left), 壹(One)" | *Wang et al. (2021)* | 8 (6 m and 2 f/age from 22–27) | SynAmps 2 | Average: 90 |

**Note:**
A, authors; M, methods; P, pronunciation materials; D, datasets; S, subjects (number); DE, device; E, the evaluation indicators (accuracy: %); m, males; f, females; TDC, time domain characteristics; LC, linear classifier; WD, wavelet decomposition; HON-ICA, higher orders of Non-Gaussianity independent component analysis; CM, covariance matrix; SCM, spatial covariance matrix; RVM, relevance vector machines; ERP, event-related potential; RLDA, regularized linear discriminant analysis; PSD, power spectral density; SampEn, sample entropy; ELM, Extreme Learning Machine; WPD, wavelet packet decomposition; LightGBM, light gradient boosting machine.

*(2021)* used CNNs to classify brain spatial features formed by EEG channels and predict user states. However, this method was not compared with others, making it impossible to assess its relative performance. *Ahn et al. (2022)* modified the convolutional layers of CNNs by replacing single convolution kernels with multi-scale kernels, allowing for the extraction of features from different frequency bands and showing high and stable classification performance across different datasets. *Kwon et al. (2023)* proposed an improved deep learning model by incorporating multi-layer perceptrons (MLP) into ShallowConvNet and EEGNet to enhance cross-paradigm subject identification capabilities in BCI systems. Table 5 presents some applications of CNNs in SI-BCI neural decoding.

RNNs are used to process variable-length sequential data, such as time-series data and sound. They consist of a series of connected feedforward neural networks. The system uses temporal correlations to represent input history and predict outputs within the network. Figure 10 shows a typical recurrent neural network structure. Research on this type of network primarily focuses on the classification and prediction of continuously varying signals over time. *Chengaiyan, Retnapandian & Anandan (2020)* first used brain

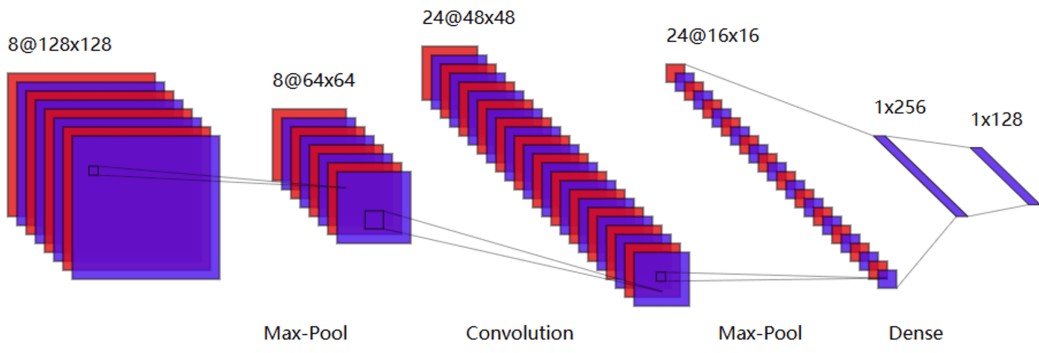

**Figure 9 Typical convolutional neural network model.**

connectivity estimators and RNNs to identify vowels, achieving a classification accuracy of 72%, which demonstrated the effectiveness of this method. *Retnapandian & Anandan (2023)* proposed a method based on EEG subband signal features and the RNN model. They decomposed the EEG signal into five subband frequency bands and extracted energy coefficients (RMS, mean absolute value (MAV), integrated EEG (IEEG), simple square integral (SSI), variance of EEG (VAR), average amplitude change (AAC)) and relative power from each subband as features, using the RNN for multi-class vowel classification. After multiple rounds of experiments, the classification accuracy approached 90%. However, the training of the RNN uses the backpropagation through time (BPTT) algorithm. Due to difficulties with gradient explosion and vanishing, backpropagating gradients over long time intervals is challenging (*Gomez et al., 2017*; *Vorontsov et al., 2017*). Long short-term memory (LSTM) and gated recurrent units (GRU) have become popular alternatives to RNNs. *Jeong et al. (2022)* used GRUs to decode spatio-temporal frequency features at the sentence level. *Hernandez-Galvan, Ramirez-Alonso & Ramirez-Quintana (2023)* employed 1D convolution layers and two layers of bidirectional GRUs to extract time-frequency features from EEG signals, using prototypical networks for classification. The average accuracy across two public datasets was above 91%. Table 6 presents the applications of RNNs and their variants.

CapsNets aim to address the challenges faced by CNNs in handling spatial hierarchies and pose variations in images. Its core advantage lies in its ability to capture the hierarchical structure and pose variations of objects within images, providing a more refined and robust feature representation than CNNs. Figure 11 shows a simple capsule network model (*Aboussaleh et al., 2024*). *Macías-Macías et al. (2023)* proposed a capsule network-based classification model, CapsK-SI, which achieved recognition of bilabial sounds, nasal sounds, consonant-vowel combinations, and the /iy/ and /uw/ vowels. It also generated relevant brain activity maps, offering clues for further understanding the neural mechanisms of speech imagination. *Ramirez-Quintana et al. (2023)* proposed a deep capsule network model called CapsVI, which achieved an average accuracy of 93.32% in paired classification. Table 7 presents applications of CapsNets and its variants.

Research on other deep learning methods has also demonstrated their effectiveness in decoding SI-BCI systems. *Saji et al. (2020)* and *Watanabe et al. (2020)* respectively

**Table 5 Application of convolutional neural networks in SI-BCI neural decoding.**

| A | M | P | D | S | DE | E |
|---|---|---|---|---|---|---|
| *Saha & Fels (2019)* | HDNN | "/a/, /i/, /u/"<br>"in, out, up"<br>"cooperate, independent" | *Nguyen, Karavas & Artemiadis (2017)* | 15 (11 m and 4 f/age from 22–32) | BrainProducts ActiCHamp amplifier | 71.1–90.7 |
| *Saha, Fels & Abdul-Mageed (2019)* | CNN+LSTM | Vowels, words | Kara One (*Zhao & Rudzicz, 2015*) | 12 (8 m and 4 f/age mean 27.4) | SynAmps RT | Highest: 85.23 |
| *Macías-Macías et al. (2020)* | CSP<br><br>sCNN/sCapsNet | /a/,/u/ | *DaSalla et al. (2009)* | 3 (2 m and 1 f/age from 26–29) | BioSemi ActiveTwo | sCapsNet 71.9<br>sCNN 67.63 |
| *Lee et al. (2021)* | LFSICF | "hello, help me, thank you"<br>"stop, yes" | BCI Competition V3 (*BCI Competition Committee, 2022*) | 15 (age from 26–29) | N/A | Average: 59.47 |
| *Park, Yeom & Sim (2021)* | MI+CNN | "hello" | Private data | 10 (5 m and 5 f/age from 20–30) | Compumedics 64-channel EEG and STIM2 | Average: 88.25 ± 2.34 |
| *Wang & Wang (2022)* | MBBNPCNN | Chinese characters "左 (Left), 壹(One)" | *Wang et al. (2021)* | 10 (7 m and 3 f/age from 22–28) | SynAmps 2 | Average: 83.72 |
| *Ahn et al. (2022)* | MSCT | "go, stop, in, cooperate" | 1. Private data | 1.40 (40 m /age from 22–28) | BrainAmp | Private dataset 62 |
| | | | 2. ASU (*Nguyen, Karavas & Artemiadis, 2017*) | 2.15(11 m and 4 f/age from 22–32) | BrainProducts ActiCHamp amplifier | ASU 72 |
| *Cui et al. (2023)* | TDSC, STFT, DWT, CSP Deep ConvNet | "/a/, /u/, /i/, /ü/"<br>"/b_/, /f_/, /j_/, /l_/, /m_/" | *Li, Pun & Chen (2021)* | 11(7 m and 4 f/age from 20–30) | 64-channel electrode cap | Highest: 68.7 |
| *Jeong et al. (2022)* | CNN<br>GRU | "I, partner, move, have, drink, box, cup phone" | Private data | 11 (6 m and 5 f/age from 20–34) | BrainVision recorder | Percent Valid Correct: 81 |
| *Kwon et al. (2023)* | ShallowConvNet<br><br>EEGNet<br>MLP | Word | Private data | 5 (2 m and 3 f/age from 23–26) | BrainProducts' actiCham | 34–50 |
| *Nitta et al. (2023)* | LPA<br>PCA<br>CNN | Japanese syllables | Private data | 1 (1 f/age 23) | g.HIAMP | Average: 72.6 |

**Note:**
A, authors; M, model; P, pronunciation materials; D, datasets; S, subjects (number); DE, device; E, evaluation indicators (accuracy: %); m, males; f, females; HDNN, hybrid deep neural network; CNN, convolutional neural network; LSTM, long short-term memory; LFSICF, length first speech imagination classification framework; MI, mutual information; MBBNPCNN, parallel convolutional neural network based on multi-band brain networks; MSCT, multiscale convolutional transformer; TDSC, time domain statistical characteristics; STFT, short-time Fourier transform; DWT, discrete wavelet transform; GRU, gated recurrent unit; MLP, multi-layer perceptron; LPA, linear predictive analysis; PCA, principal component analysis.

employed DWT and speech amplitude envelope rhythm extraction to extract EEG features, using different classifiers for speech imagination classification, demonstrating the application potential of deep learning models in different linguistic contexts. *Islam & Shuvo (2019)* proposed a deep learning-based improvement method for beta-band selection, combining DenseNet and Gramian Angular Field to address the issue of EEG

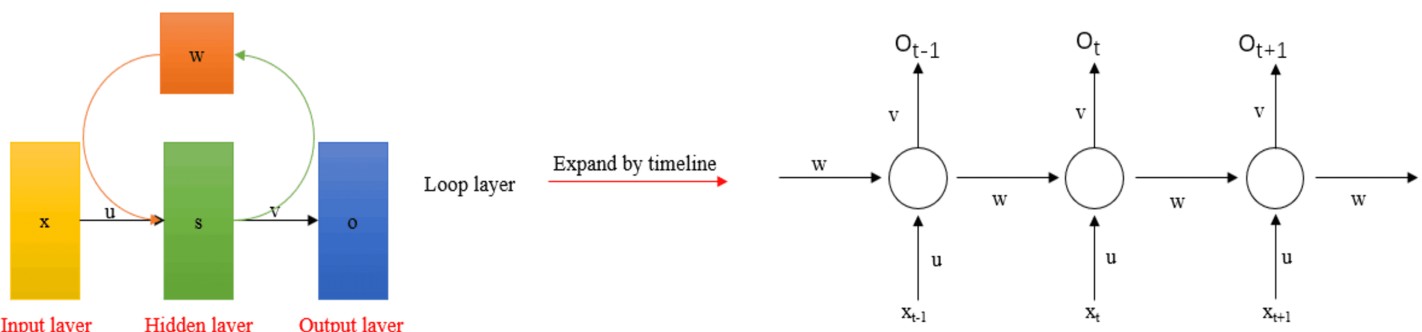

**Figure 10  Typical recurrent neural network model.**               

**Table 6  Application of recurrent neural networks in SI-BCI neural decoding.**

| A | M | P | D | S | DE | E |
|---|---|---|---|---|---|---|
| *Chengaiyan, Retnapandian & Anandan (2020)* | BCE, RNN, DBN | CVC words | Private data | 6 ( 3 m and 3 f/age mean 20) | Placing Ag/AgCl electrodes | 80 |
| *Hernandez-Galvan, Ramirez-Alonso & Ramirez-Quintana (2023)* | Proto-Speech | Vowels, short words, long words | 1. Kara One (*Zhao & Rudzicz, 2015*) | 1. 3 (2 m and 1 f/age from 26–29) | SynAmps RT | Kara One: 99.89–99.92 (Binary classification) |
| | | | 2. ASU (*Nguyen, Karavas & Artemiadis, 2017*) | 2. 15(11 m and 4 f/age from 22–32) | BrainProducts ActiCHamp amplifier | 91.51(Multi classification) ASU: 93.70 (Multi classification) |
| *Jeong et al. (2022)* | CNN, GRU | "I, partner, move, have, drink, box, cup phone" | Private data | 11 (6 m and 5 f/age from 20–34) | BrainVision Recorder | Percent valid correct: 81 |
| *Retnapandian & Anandan (2023)* | RNN | /a/, /e/, /i/, /o/, /u/ | Private data | 5 (5 m) | Wireless Emotiv EPOC+ Neuro-technology | 84.5–88.9 |

**Note:**
A, authors; M, model; P, pronunciation materials; D, datasets; S, subjects (number); DE, device; E, evaluation indicators (accuracy: %); m, males; f, females; BCE, brain connectivity estimators; DBN, deep belief networks; CVC, consonant–vowel–consonant; RNN, recurrent neural network; CNN, convolutional neural network; GRU, gated recurrent unit.

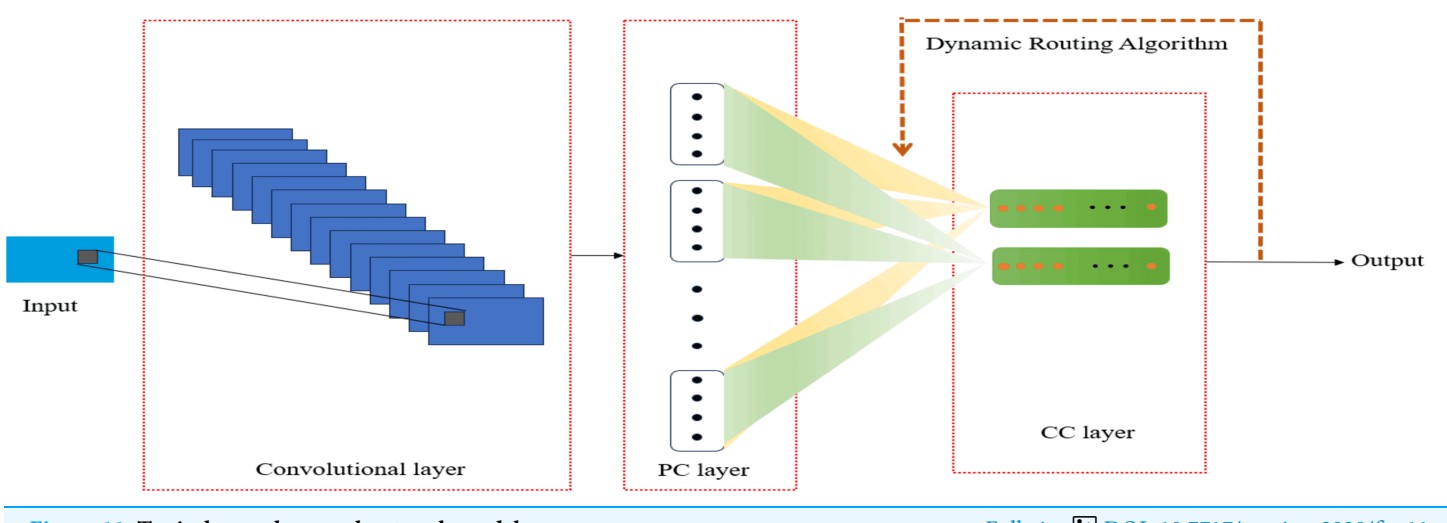

**Figure 11  Typical capsule neural network model.**               

**Table 7 Application of capsule neural network in SI-BCI neural decoding.**

| A | M | P | D | S | DE | E |
|---|---|---|---|---|---|---|
| *Macías-Macías et al. (2023)* | CapsK-SI | Phonemes, words | Kara One (*Zhao & Rudzicz, 2015*) | 12 (8 m and 4 f/age mean 27.4) | SynAmps RT | 89.70–94.33 |
| *Ramirez-Quintana et al. (2023)* | CSP, CapsVI | /a/, /u/, /no/ | *DaSalla et al. (2009)* | 3 (2 m and 1 f/age from 26–29) | BioSemi ActiveTwo | Average: 93.32<br>Highest: 94.68 |

**Note:**

A, authors; M, model; P, pronunciation materials; D, datasets; S, subjects (number); DE, device; E, evaluation indicators (accuracy: %); m, males; f, females; CSP, common spatial pattern.

signal classification accuracy, significantly improving classification accuracy. *Panachakel & Ganesan (2021)* sed a transfer learning model based on ResNet50 to classify the ASU imagined speech EEG dataset, showcasing the effectiveness of transfer learning in EEG signal classification. Table 8 presents the applications of some other deep learning algorithms.

## Algorithm comparison and evaluation

The compilation and analysis of existing SI-BCI neural decoding algorithms are summarized in Fig. 12. The following trends emerge from a quantitative analysis: Machine learning methods predominantly utilize SVM, which demonstrate stable performance and high classification accuracy, making them the baseline method in this field and accounting for the majority of applications. Among deep learning algorithms, CNNs and their variants have demonstrated exceptional performance.

Model performance evaluation involves various metrics, such as accuracy, precision, recall, F1-score, confusion matrix, receiver operating characteristic (ROC) curves, and area under the curve (AUC). Each metric emphasizes different aspects of model performance, making the selection of appropriate evaluation criteria critical. In the field of SI-BCI, average accuracy is the primary performance metric for evaluating SI-BCI algorithms. Additionally, metrics such as recall, precision, and F1-score are often included to offer a more comprehensive assessment of model performance. Thus, accuracy is chosen as the main evaluation metric in this study to facilitate clear comparison of the relative performance of various models. For both binary and multi-class classification tasks, accuracy is defined as the proportion of correctly classified samples to the total number of samples. However, in multi-class classification, accuracy calculations must account for multiple categories, which makes comparisons between binary and multi-class tasks more nuanced. Table 9 presents the highest accuracy achieved by different models and methods.

## CHALLENGES AND PROSPECTS

A comprehensive literature review reveals that EEG-based encoding and decoding research for SI-BCI has made significant advancements in recent decades. Ongoing studies on brain region physiology, paradigm design, and decoding algorithm innovations are expected to gradually overcome challenges, such as unnatural SI-BCI brain-machine interactions. However, several unresolved issues remain and require further exploration.

**Table 8 Application of other deep learning algorithms in SI-BCI neural decoding.**

| A | M | P | D | S | DE | E |
|---|---|---|---|---|---|---|
| *Islam & Shuvo (2019)* | DenseNet, GAF | Syllables | Kara One (*Zhao & Rudzicz, 2015*) | 12 (8 m and 4 f/age mean 27.4) | SynAmps RT | DenseNet: 90.68 |
| | | Words | | | | GASF: 90.54 |
| *Saji et al. (2020)* | DWT, LDA/SVM/ KNN | /a/, /u/ | Public dataset (*ATR brainliner, 2015*) | 3 (8 m and 4 f/age from 26–29) | N/A | 72–80 |
| *Watanabe et al. (2020)* | EEGOSESM | Japanese syllables [ba], [ba:] | Private data | 18 (12 m and 6 f/age mean 23.8) | BrainAmp | 54.7 (Perceived speech) |
| | | | | | | 38.5 (Imagining speech) |
| *Panachakel & Ganesan (2021)* | 3DCE, RTL | Long words | ASU (*Nguyen, Karavas & Artemiadis, 2017*) | 15 (11 m and 4 f/age from 22–32) | BrainProducts ActiCHamp amplifier | H1 93.4 |
| | | Short words | | | | H2 93.1 |
| | | Vowels | | | | H3 79.7 |
| | | Short long | | | | H4 95.5 |
| | | Words | | | | |
| *Panachakel & Ramakrishnan (2021)* | MPC, Shallow neural network | Syllables | Kara One (*Zhao & Rudzicz, 2015*) | 12 (8 m and 4 f/age mean 27.4) | SynAmps RT | Average: 75 |
| *Naebi & Feng (2023)* | LSIM | M, A | Private data | 3 (age from 30–40) | SynAmps | 55–98 |
| | NCS | | | | | |
| | LBGC | | | | | |
| | DFD | | | | | |
| *Pan et al. (2024)* | WST, KPCA, XGBoost | /ka:m/,/kwest/, /piəriə/, /əpas/,/spei/ | Private data | 7 (age from 22–46) | Emotiv EPOC Flex | 78.73 |
| *Wu et al. (2024)* | PSD, LDA | /fɔ/, /gi/ | 1. *Bhadra, Giraud & Marchesotti (2025)* | 1. 15 (10 m and 5 f/age from 19–29) | ANT Neuro system | 62.2 |
| | | | 2. Private data | 2. 20 (7 m and 13 f/age from 20–30) | | |

**Note:**
A, authors; M, model; P, pronunciation materials; D, datasets; S, subjects (number); DE, device; E, evaluation indicators (accuracy: %); m, males; f, females; GAF, Gramian Angular Field; DWT, discrete wavelet transformation; EEGOSESM, EEG oscillation and speech envelope synchronization model; 3DCE, 3D compression encoding; RTL, ResNet50 transfer learning; MPC, mean phase coherence; LSIM, lip-sync imagery model; NCS, new combinations of signals; LBGC, linear bond graph classifier; DFD, deep formula detection; WST, wavelet scattering transform; KPCA, kernel principal component analysis; XGBoost, Extreme Gradient Boosting; PSD, power spectral density; LDA, linear discriminant analysis; H1, highest long words average accuracy; H2, highest short words average accuracy; H3, highest vowels average accuracy; H4, highest short long words average accuracy.

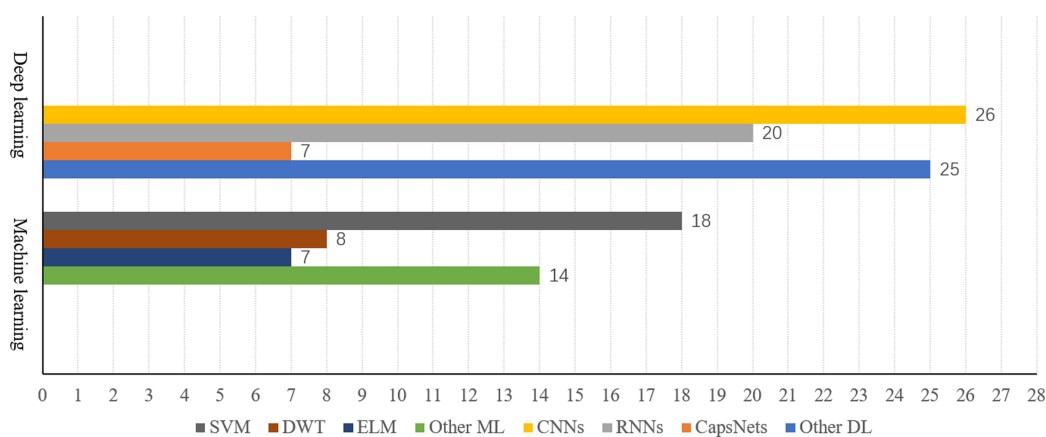

**Figure 12 Comparison of the number of machine learning algorithms and deep learning algorithms.**

**Table 9 The optimal accuracy among various models.**

**Binary classification**

| B | M | E |
|---|---|---|
| CSP | EMBCSP | 89.42 |
| SVM | | |
| CSP | STWFBCSP | 84.87 |
| ELM | | |
| MFCCs | SVM | 88.3 |
| SVM | DTW | |
| CSP | WD | 81.25–98.75 |
| RVM | LC | |
| CNN | MBBNPCNN | 83.72 |
| RNN | Proto-Speech | Kara One 99.89–99.92 |
| CNN | CapsNet-VI | 93.32 |
| CNN | | H1 93.4 |
| CSP | 3DCE | H2 93.1 |
| SVM | RTL | H3 79.7 |
| | | H4 95.5 |

**Multi classification**

| B | M | E |
|---|---|---|
| CSP | CSP, CCF, | 79.33–88.26 |
| SVM | SVM | |
| CSP | CM | 70 |
| SVM | RVM | |
| CNN | LFSICF | 59.47 |
| RNN | Proto-Speech | Kara One 91.51 |
| | | ASU 93.70 |

Note:
EMBCSP, evidential multi-band common spatial pattern; STWFBCSP, sub-time window filter bank common spatial pattern; DTW, dynamic time warping; WD, wavelet decomposition; LC, linear classifier; MBBNPCNN, parallel convolutional neural network based on multi-band brain networks; 3DCE, 3D compression encoding; RTL, ResNet50 transfer learning; CCF, cross-correlation function; B, baseline (model); M, methods (model); E, Evaluation indicators (accuracy: %); H1, highest long words average accuracy; H2, highest short words average accuracy; H3, highest vowels average accuracy; H4, highest short long words average accuracy; LFSICF, length first speech imagination classification framework; CM, covariance matrix; CNN, convolutional neural network; SVM, support vector machine; CSP, common spatial pattern, ELM, Extreme Learning Machine; MFCCs, Mel-frequency cepstral coefficients; RVM, relevance vector machine; RNN, recurrent neural network.

To guide future research on SI-BCI systems, we summarize the current challenges and propose potential research directions.

## Difficulty in acquiring high-quality signals and the principles of brain region connectivity in speech processing

**Challenges:** Our analysis of Tables 1–8 indicates that most studies use EEG acquisition devices with 32 or more channels. However, signal quality is often influenced by participant characteristics, including age, gender, and handedness. Furthermore, prolonged experimental sessions may lead to participant fatigue, impacting attention and

reaction speed, which increases EEG variability and noise artifacts (*e.g.*, ocular and electromyographic artifacts), ultimately reducing signal quality (*Zhang, Li & Chen, 2020*; *Borirakarawin & Punsawad, 2023*; *Kim, Lee & Lee, 2020*; *Lee et al., 2021*). Additionally, the experimental environment plays a crucial role. Prolonged recording sessions increase the risk of environmental interference during signal acquisition. For instance, *Jafferson et al. (2021)* reported an 18% error rate in classification results due to electrical interference during data collection. In "SI-BCISI-BCI Paradigm", we reviewed recent advancements in brain connectivity studies. Broca's and Wernicke's areas are known to be essential in speech processing—Broca's area is responsible for speech production and articulation, whereas Wernicke's area facilitates speech comprehension. However, many existing studies collect EEG signals from the entire brain without focusing on electrode placement in these critical speech-related regions (*Wang et al., 2021*; *Sikdar, Roy & Mahadevappa, 2018*; *Zhao, Liu & Gao, 2021*; *Pan et al., 2024*; *Wang et al., 2020*).

**Future Directions:** To advance the understanding of speech-related brain mechanisms, the following research directions are proposed: Interdisciplinary research–Strengthen collaboration with fields such as anatomy and neuroscience to conduct more precise structural and functional analyses of speech-related brain regions, particularly Broca's and Wernicke's areas. This will enable deeper investigation into the neural signal transmission and cortical activation mechanisms underlying speech intention, potentially uncovering functional micro-units within these key linguistic centers (*Zhang, Guo & Chen, 2023*). Optimization of EEG acquisition equipment–Design and refine more efficient and sensitive EEG acquisition devices capable of capturing a broader range of neural activity frequencies. Enhancing the precision of EEG recordings will improve the detection of subtle neural activities involved in speech generation. These research directions hold promise for revealing the high-level cognitive functions underlying speech generation and comprehension and advancing the development of high-sensitivity EEG systems for applications in neurolinguistics.

## Innovation in SI-BCI paradigms: limited focus on words and sentences

**Challenges:** As summarized in "SI-BCI Neural Encoding" and Tables 1–8, vowel imagery has been extensively studied (*Chengaiyan & Anandan, 2022*; *DaSalla et al., 2009*; *Cooney, Folli & Coyle, 2018*; *Wu & Chen, 2020*; *Agarwal et al., 2020*), whereas research on short words, long words, and phrases—such as English words and Chinese characters—is relatively scarce (*Alizadeh & Omranpour, 2023*; *Kamble, Ghare & Kumar, 2022*; *Sereshkeh et al., 2017*; *Panachakel & Ramakrishnan, 2022*). This limitation may stem from the design of existing paradigms, as a single paradigm may not be sufficient to fully address complex speech imagery tasks.

**Future Directions:** To innovate SI-BCI paradigms, future research should focus on designing paradigms that incorporate word-, character-, phrase-, and sentence-level speech intention tasks. Additionally, integrating multimodal signals (*e.g.*, electromyography (EMG), eye movement signals) and multimodal paradigms (*e.g.*, motor imagery and external stimuli) could improve the naturalness and usability of speech

intention paradigms for real-world applications. *Wang et al. (2019, 2022)* proposed a hybrid BCI concept by designing a three-class task system that combines motor imagery and speech imagery, leading to improved classification accuracy. This suggests that multimodal data fusion can significantly enhance the efficiency and precision of BCI systems. *Tong et al. (2023)* further explored multimodal speech imagery applications by integrating silent reading and handwriting imagery, which elicited stronger EEG features compared to traditional motor imagery paradigms, thereby improving classification accuracy. *Naebi & Feng (2023)* proposed a novel communication imagery model as an alternative to speech-dependent mental tasks. They introduced a lip imagery model, achieving classification results ranging from 55% to 98%, with the highest performance achieved through a novel signal combination approach. Notably, *Silva et al. (2024)* leveraged BCI and natural language processing (NLP) technologies to enable a patient to communicate in Spanish and English. This groundbreaking study may provide new insights into the development of bilingual BCIs.

## Excessive signal preprocessing and lack of multimodal data fusion in SI-BCI neural encoding

**Challenges:** Some studies have applied Autoencoder layers to handle incomplete and noisy EEG signals (*Zhang et al., 2018*; *Ali, Mumtaz & Maqsood, 2023*), while others have employed noise-assisted multivariate empirical mode decomposition (NA-MEMD) for signal processing (*Park & Lee, 2023*). However, these methods significantly increase preprocessing time, hindering real-time applications. *He et al. (2023)* highlighted that relying solely on EEG signals for unimodal data processing limits the model's ability to capture the complexity of brain activity.

**Future Directions:** Beyond EEG, other physiological signals—such as fNIRS, electrooculography (EOG), and EMG—should be integrated into SI-BCI neural encoding for imagined speech decoding. Although most research focuses on single-modal signals, multimodal classification has emerged as a promising direction (*Wang et al., 2023*). *Guo & Chen (2022)* developed an fNIRS-based 4-class speech imagery BCI, achieving >70% mean subject-specific accuracy in binary-class settings (all pairwise comparisons between two vowels) across 0–2.5 s and 0–10 s time windows. *Vorreuther et al. (2023)* and *Rezazadeh Sereshkeh et al. (2019)* proposed a multi-class hybrid fNIRS-EEG BCI for imagined speech, demonstrating significantly higher accuracy than unimodal approaches. These findings highlight the complementary nature of diverse neural signals in enhancing BCI performance. Future studies should adopt mobile, wearable sensors to collect multimodal physiological data (facial expressions, EEG, *etc.*) and fuse them within a unified framework to improve online real-time feature extraction and imagined speech recognition accuracy.

## Limited SI-BCI datasets and overreliance on binary classification tasks

**Challenges:** In Tables 1–8 and "Difficulty in Acquiring High-Quality Signals and the Principles of Brain Region Connectivity in Speech Processing", we analyzed available datasets, revealing that most studies rely on private datasets with fewer than ten

participants. This poses a significant challenge for neural decoding, as the lack of large datasets prevents deep learning models from leveraging large-scale data analysis, thus affecting model performance. Additionally, most studies still focus on binary classification tasks, limiting their applicability to real-world BCI applications. Furthermore, the absence of diverse evaluation metrics hinders comprehensive performance assessment.

**Future Directions:** In SI-BCI neural decoding, it is crucial to continue advancing deep learning algorithms, particularly CNN-based models, to enhance the capability of handling complex, nonlinear transformations. Expanding research from binary classification to multiclass classification is essential. For example, *Hernandez-Galvan, Ramirez-Alonso & Ramirez-Quintana (2023)* achieved high accuracy across multiple classes using two public datasets, demonstrating the feasibility of multiclass SI-BCI tasks. Current public datasets are predominantly vowel-based, such as Kara One and AUS. Therefore, developing high-quality word- and sentence-level public datasets is a critical direction for future SI-BCI advancements. A recent contribution in this area is ChineseEEG, a high-density EEG dataset compiled by *Mou et al. (2024)*, containing approximately 13 h of Chinese text reading EEG recordings from 10 participants. This dataset fills a significant gap in Chinese language EEG datasets and represents a major step toward improving linguistic diversity in SI-BCI research.

## CONCLUSIONS

In recent years, with the rapid advancement of brain-computer interface (BCI) technology, research on speech imagery decoding based on electroencephalography (EEG) has garnered significant attention. This article systematically reviews the latest progress in EEG-based speech imagery BCI decoding, focusing on key research findings and challenges in brain region connectivity, SI-BCI experimental paradigms, encoding techniques, and decoding algorithms. Through an analysis of existing studies, we suggest that future research should further emphasize brain region mechanisms, paradigm innovation, encoding principles, and decoding algorithms to advance the practical application of speech imagery BCI. With ongoing technological advancements, personalized and miniaturized speech intention BCI systems are expected to emerge. In the medical field, speech imagery BCI can assist aphasia patients in restoring communication abilities and improving their quality of life. In non-medical domains, speech imagery BCI can be applied to human-computer interaction and smart home systems, providing a more natural and efficient interaction modality. We hope this article will inspire further innovation and development in BCI paradigms and neural encoding-decoding research.

### Funding

This research was funded by the National Natural Science Foundation of China (No. 51405252), the China Postdoctoral Science Foundation of China, grant number (No.

2020M672101), the International Cooperation Fund of Qilu University of Technology of China, grant number (No. QLUTGJHZ2018022) and the Innovation Team Fund of Qilu University of Technology, and the Cooperative Innovation Fund of Qilu University of Technology of China (No. 2021CXY-04). The funders had no role in study design, data collection and analysis, decision to publish, or preparation of the manuscript.

## Grant Disclosures

The following grant information was disclosed by the authors:
National Natural Science Foundation of China: 51405252.
China Postdoctoral Science Foundation of China: 2020M672101.
Qilu University of Technology of China: QLUTGJHZ2018022 and 2021CXY-04.

## Competing Interests

The authors declare that they have no competing interests.

## Author Contributions

- Ke Su conceived and designed the experiments, prepared figures and/or tables, authored or reviewed drafts of the article, and approved the final draft.
- Liang Tian conceived and designed the experiments, performed the experiments, analyzed the data, performed the computation work, prepared figures and/or tables, authored or reviewed drafts of the article, and approved the final draft.

## Data Availability

This is a literature review.

## Supplemental Information

Supplemental information for this article can be found online at http://dx.doi.org/10.7717/peerj-cs.2938#supplemental-information.

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
