# Peer review of "Systematic review: progress in EEG-based speech imagery brain-computer interface decoding and encoding research"

_PeerJ Computer Science, doi:10.7717/peerj-cs.2938_

## Round 0.1 · original submission · Major Revisions

The content of the article is interesting; however, the reviewers identified several issues that need to be carefully examined and addressed by the authors. Please consider all the suggestions and comments from the reviewers and prepare a revised version of the article accordingly.

Reviewer 1 ·

Basic reporting

This paper provides a comprehensive overview on different EEG-based speech imagery BCIs with an extensive literature research and summary on SI-BCI neural encoding and decoding techniques. The work tries to cluster related literature into differences in paradigms, encoding and decoding techniques, thereby addressing an important issue highly relevant to the field.
However, the part regarding the BCI paradigm, is only sparsely mentioned and not clearly worked out. This should probably be presented in the paragraph on “physiological mechanism of SI-BCI system: connectivity of brain regions” which is interesting and relevant, but it stays beyond what could/should be elaborate on, a comprehensive overview on different SI-BCI paradigms. There are differences in cue presentation (auditory, visually), single vs continuous repetitions of the words, vowels, sentences, and also in imagined speech production (inner speech, envisioned speech, …) which all have an influence on neural encoding/decoding, while there is not really a clear separation between them.
This major issue in the field of SI-BCI research is only presented sparsely and further only mentioned in the neural encoding section which raises questions whether neural encoding and BCI paradigms can be clearly separated at all. The paragraph on neural encoding (line 216) even starts with listing several paradigms, while the previously mentioned paradigm paragraph only reports on connectivity.
Although the paper provides an interesting contribution by looking at Speech Imagery BCI from the perspective of neural decoding and encoding, the encoding part has more potential, e.g. to highlight current bottlenecks and pitfalls in study design and BCI paradigm approaches.
In the current state this work presents an extension of Panachakel’s review paper from 2021 [15] which mainly focused on the decoding side but the unique additional contribution is not clearly stated.
The main points of concern speaking for a major revision are listed briefly in the following and in more detail later in additional comments:
- Structure and separation of encoding and paradigm not clear
- Differences and details regarding SI BCI paradigms not clearly worked out
- Better highlight difference and additional contributions in comparison to existing literature reviews, e.g. Panachakel.
- Missing relevant information on study details in the tables (nr. participants, recording device, …)
- Figures of insufficient quality and information value

Experimental design

The survey methodology followed the PRISMA method which seems adequate for the purpose of the paper and is sufficiently referenced and explained throughout the text and illustrated in figure 2. Quantity and quality of identified related works is good and well cited, except one small part mentioned in general comments. The organization and structure of the paper should be improved, there are some issues with the separation in paradigm and encoding, as mentioned in detail in basic reporting and the additional comments.

Validity of the findings

The conclusion identifies unresolved questions and future directions, however partially a bit short especially regarding SI-BCI paradigms and encoding principles. As an example, it is suggested focusing on words and sentences, but it is not further motivated why and what potential benefits over other strategies could be. The encoding part provides this motivation/justification and sufficiently reports on potential further directions.

Additional comments

Figures:
- Overall too many figures with insufficient information value
- Several of them could be merged into one figure without being overly complex, e.g. 3,5, 6/9 could be merged to one overview figure highlighting paradigm, encoding, decoding and their relationship
- Figure 6 and 9 are called the same: The SI-BCI Neural Decoding Process, figure 9 is not referenced in the text, so I guess it should be excluded although I prefer it over figure 6, maybe consider replacing 6 with 9
- Figure 2 mentions neural coding, does this include encoding and decoding?
- Figure 4 appears squeezed
- Figure 10, what is other? In the text you mention more methods, e.g. classifiers (RF, KNN, ELM…) and features (DWT, MFCCs,…). I would recommend referring to the tables or figure 14 and delete figure 10, because it does not add any value imo.

Detailed remarks:
- Number of participants is an interesting factor which should be mentioned in the tables and overviews.
- Same holds for recording devices, some record with consumer headsets while others use clinical grade headsets with much higher resolution and signal quality.
- Panachakel et al. [15] are mentioned with their review paper but it is not presented as such. I suggest presenting this work as review paper and precisely point out the differences to your work and in how far you extend, or what is different in your methodology.
- Line 52, there is actually work showing that EEG signals recorded during imagined and spoken speech are similar enough to transfer classifiers in between them which could be mentioned (DOI:10.1109/SMC53654.2022.9945447). Title says silent speech but it actually is imagined, that’s probably why you did not find it in the first place. A follow up paper on the techniques makes it clearer (DOI: 10.1145/3656650.3656654)

- Line 106 to 112, This paragraph is a bit confusing though being essential to understand the scope of your work. To me, BCI paradigm is a part of the neural encoding or does BCI paradigm include encoding and decoding and if so, please make it clear, maybe a reference to fig 4 might be helpful.
- Line 109, I guess that “paradigms and neural decoding” should be “encoding”
- Figure 3, I am not sure if I agree with the circular structure. I agree that without a valid paradigm, we cannot get to a proper encoding and following decoding. But in how far does the decoding influence the BCI paradigm? It might be that without a proper decoding we do not get to a satisfying classification and therefore performance of the system which affects the paradigm itself by user performance. Imo just writing that paradigms cannot be “validated” (line 111) is not sufficient and should be further elaborated on.
- Line 121 “Coding” should be replaced with “Encoding”
- Line 128, I suggest mentioning features instead of the phenomena you are observing. In my opinion ERP is not a feature but a whole principle which looks at a specific form of the signal in the time domain correlated to a stimulus, but as a feature you are measuring the amplitude of the signal. Same holds for ERD/ERS, you are measuring the power decrease/increase of the signal as a feature while the desynchronization describes the development of the power over time that you are investigating as ERD/ERS. Better use Discrete Wavelet Transform as an example. And for Broca and Wernicke I suggest replacing it with CSP.
- Line 139 – 148, not a single reference, please add at least one basic reference for Broca Wernice, connectivity in encoding and decoding
- Line 216 ff. at this point I am confused if you mean neural encoding or speaking about the BCI paradigm. The type of speech to repeat, e.g. vowel or word is imo clearly part of the BCI paradigm. In line 218 you even mention that it can be divided into three paradigms. I would highly recommend reconsidering the separation into paradigm and encoding and rather go with encoding and decoding only. Or make the paradigm include the tasks and limit encoding solely to feature extraction, which does not make much sense imo.
- Line 227, multimodal combination seems a bit off to me. There are different paradigms/techniques how to produce imagined speech. You can envision speaking while focusing only on imagined facial muscle movement, you can imagine hearing yourself/someone speaking (inner speech) you can imagine seeing yourself/someone speaking or imagine seeing the word/text to speak (envisioned speech). This PhD thesis (DOI: 20.500.11880/35881) chapter 2.4 gives a broad overview of different types of imagined speech and paradigms that might be of interest in this case.

·

Basic reporting

The authors did great job in presenting the article “Progress in EEG-based speech imagery brain 1 computer interface decoding and encoding researc.” The only concern I have is about including references that was published more than 10 years ago, while modern research is available in the same area especially when presenting machine learning techniques such as SVM, LSTM, and CNN that are used for classifying imagined speech. This includes but not limited to [40], [87], [96]… etc. Citing the most state-of-the-art research articles is essential to add value to this work.

Experimental design

No comments.

Validity of the findings

No comments

Additional comments

No comments

Reviewer 3 ·

Basic reporting

1– The review is in the journal scope and focuses on a hot topic in the field of EEG-based brain-computer interfaces, that is speech imagery. In particular, the Authors want to provide a literature overview on the speech imagery paradigms, the neural encoding, and decoding.
The Authors follow the PRISMA guidelines to prepare their review.
a– While the introductory parts on BCI and speech imagery are very clear and provide a good amount of information to understand the topic of interest, the manuscript presents a difficult structure to follow and some important background information on the three main points of interest seem to be reported in different sections.
b– While there is an initial assessment of the PRISMA guidelines, some important details on the review process are missing.
c– Moreover, the aims of the Authors could be further explained in the introduction, that should also report a clear understanding of the differences between the proposed review and the other published literature surveys.

Experimental design

2– I think the paper requires a restructuring in terms of sections, which are currently not clearly divided and may confuse the reader.
a– I suggest using the Introduction to clearly set the basic information to understand the topic, what is currently in the literature, and what are the Authors’ purposes for this review. This would comprise the subsections present in pages 1 and 2, “Brain Computer Interface” and “Speech Imagery” that I think can be easily merged together. For further details on how to improve the introduction, please refer to comment 3.
b– It is very good that the Authors exploit the PRISMA guidelines, however I think they should expand the section related to the “Survey methodology” to better highlight the paper collection and selection process. Please, refer to the PRISMA guideline for further details as well as exploit some of the review examples reported in comment 3 for reference. Comment 4 contains further suggestions on the content of the “Survey methodology” section.
c– Considering the current structure of the paper, it is not clear what is the core of your review. I suggest providing a clear map to navigate the various sections and subsections and let the reader understand why you divide the paper in specific sections.
d– The section “Relationship between the Paradigm, Neural Encoding, and Neural Decoding in an SI-BCI System” has a great number of subsections that are difficult to navigate. Please, consider reducing the subsections and providing a clear understanding of the key elements that are part of your review analyses. Please, see more comments at point 5.
e– I think the datasets presentation could be moved to a specific section preceding the analysis of the papers. It would be then clearer the type of data the papers are dealing with and the classification task.

3– The introduction presents a good overview of the topic of interest, i.e., speech imagery in the field of EEG-based brain-computer interfacing.
a– I suggest anticipating some brief explanations on paradigms, neural encoding and decoding, and brain connectivity to allow any reader to better understand your survey interests and the importance they have for your review.
b– When presenting some of the literature works providing some insights on the topic (between lines 66 and 77), I think that some literature survey papers covering some of the aspects that are of interest to the Authors are missing. Reporting the differences with such articles and highlighting the key points of the present review work would give further value to the manuscript.
In what follows I provide some reading suggestions, that could help in providing a clear understanding of the point of view of your narrative and vision:
- Rahman, N., Khan, D. M., Masroor, K., Arshad, M., Rafiq, A., & Fahim, S. M. (2024). Advances in brain-computer interface for decoding speech imagery from EEG signals: a systematic review. Cognitive Neurodynamics, 1-19.
- Alzahrani, S., Banjar, H., & Mirza, R. (2024). Systematic Review of EEG-Based Imagined Speech Classification Methods. Sensors, 24(24), 8168.
- Lopez-Bernal, D., Balderas, D., Ponce, P., & Molina, A. (2022). A state-of-the-art review of EEG-based imagined speech decoding. Frontiers in human neuroscience, 16, 867281.
- Zhang, L., Zhou, Y., Gong, P., & Zhang, D. (2024). Speech imagery decoding using EEG signals and deep learning: A survey. IEEE Transactions on Cognitive and Developmental Systems.
c– As a final note, you can also consider setting a list of research questions you would like to address with your review, that you could exploit to discuss the finding of your literature search.

4– Concerning “Survey methodology” and considering following the PRISMA guidelines, I think some information is currently missing.
a– How did you use the search keywords? Did you consider combinations of the main keywords that seem to be Brain-Computer Interface, Speech Imagery, and EEG signals of speech imagery? How are deep and machine learning integrated in this search?
b– Are the inclusion and exclusion criteria fully reported? Did you consider both surveys and original research papers? Are degree theses included? Did you screen everything manually? Why did you consider starting your search from 2002?
c– How are the paradigms, neural encoding and decoding searched for? Is it something you try to figure out while screening the papers? Do the papers need to present all three aspects to be included?
d– Figure 1 seems to be missing some details. While the exclusions in the identification step are clear, the exclusions on the screening step are not clearly marked. There is also a point where it is stated that the reports are excluded for reason 1, 2, 3, etc. Please, provide a clear picture of the included and excluded works bringing to the final number of analysed papers, i.e. 110, and consider being more descriptive in the text.
e– Figure 2 is a bit confusing, considering that the main analysed aspects as declared in the abstract should be paradigms, neural encoding and decoding. Please, consider revising the figure to be more in line with the reported division or clarify this point. Reporting the percentages would also be very helpful to improve the readability of the figure and its information.

5– Section “Relationship between the Paradigm, Neural Encoding, and Neural Decoding in an SI-BCI System” is a bit confusing.
I think this section could be deputed to the background information needed to understand what follows. There are too many repetitions of the titles in different parts of the manuscript and this does not allow the reader to understand when the background information is completely presented and when the analysis of the screened papers begin. The background information seems to be scattered throughout the paper.
a– For a non-expert reader lines 106-112 may be a bit confusing not having properly introduced the terminology.
b– Leveraging on Figure 3, that could include the content of Figure 4-6, the Authors could consider reducing the subsections, presenting the interaction of the three “components” one by one.
c– Concerning the “Definition of SI-BCI paradigm” I would have expected the presentation of some of the SI-related paradigms, maybe focusing on the ones present in the surveyed papers.
d– Figure 4-6 should be revised to provide a better understanding of the presented topics. The neural encoding and decoding concepts could be better explained by introducing some examples that should be minimally presented. Encoding and decoding could also be part of the same process when using deep learning strategies, however the figures seem to be missing this point and there is some mixing between encoding and decoding. For example, feature extraction can be considered as part of the encoding step, however it seems to fall under the decoding step. Why is that so?
e– The discussion related to brain connectivity may also be integrated with the overview presentation of neural encoding and decoding, considering its very strong relation with these two mechanisms.

6– Sometimes the Authors present a dataset and do not provide a reference for it. The reference comes very later in the text as well as the description of the datasets.

7– In lines 257-263 it is stated that the previously mentioned papers have contributed to the knowledge base of SI-BCI. The Authors should better explain what they have contributed to the research field.

8– At line 362 an increase of performances is declared, however it is unclear what is the base performance.

9– At line 438 the acronym “STWFBCSP” appears, but it is never reported in its extended version. Moreover, it is stated that this algorithm has better performance compared to CSP, but there is no quantitative evidence reported to sustain this statement.

10– Figure 9 presents the process of neural decoding, however I think it needs a revision. For example, it seems that in all the cases, features are extracted in three different domains that seem also to be mapped to RGB channels. Is it always true? Why consider a color space? Moreover, some applications directly use DL to extract features from the raw data. How is this represented? Between figure and descriptive text there seem to be some contradictions.

11– Figure 10 seems a bit redundant and not extremely informative. I understand that the majority of the studies consider different combinations of CSP and SVM for the machine learning based strategies, while the deep learning ones are somewhat more varied. However, I would expect some percentages of use that would highlight the distribution of the use of these strategies among the screened papers.
Maybe the Authors could consider removing this image, having that the revision of Figure 9 and the Tables 1-8 provide more detailed information on the screened solutions.
a– Why are the datasets not reported for the machine learning related papers? I suggest adding this information for completeness and uniformity.
b– Moreover, I suggest adding the reference to the “Authors” column to have a clear link between the strategy and the paper.
c– Some evaluation indicators appear in ranges. What do the ranges represent? Moreover, stating that an average has increased without the baseline performance does not allow a complete understanding of the obtained results. Would works with no reported evaluation indicators be eligible for analysis?
d– This is also true concerning works that do not mention dataset characteristics.
Please, report the reference to the datasets also in the dedicated “Datasets” column.

12– Throughout the manuscript, sometimes it is not clear what the classification task is and the final performance achieved by a specific methodology.

13– Throughout the manuscript (tables included), some acronyms appear without presenting their extended version, such as STWFBCSP or MIBIF.

14– Summing up Figure 14, it seems that there are 54 different approaches. Are they representative of all the screened papers?

15– Table 9 reports the best accuracy of each type of method, but not providing a clear reference to the classification task does not allow a correct interpretation of the results. For example, achieving 80% accuracy for a binary task is very different from obtaining it on a multiclass problem.

16– Figure 15 also does not sum up to the final number of screened papers. Why is that so?

17– Considering that 110 papers remain after your screening process, why are there only 107 citations that comprise the background information and other survey works? All the papers would be expected to appear in your review, even in an aggregated form.

Validity of the findings

18– The final discussion of the findings derived from the screened papers should be expanded. There are many challenges not reported and the unresolved issues require a clear reference to the descriptions given in previous sections.
Some challenges may be related to the lack of standards, the heterogeneity of speech imagery paradigms, the very few reproducible papers, and so on.

19– What is it that justifies the future directions? The Authors should exploit the evidence from the paper analyses to provide their insights.

20– Please, consider adding responses to possible research questions introduced in the first section. This would better allow the readers to understand the take on messages of your work.

Additional comments

Given my previous comments, I suggest a major revision.

I think that restructuring the manuscript, providing a clear background, reporting more details on the review process (following the PRISMA guidelines and the survey examples), and being more detailed in the presentation of the screened papers (some of which seem to be missing), will greatly improve your manuscript.

---

## Round 0.2 · Minor Revisions

The authors addressed most of the comments raised by the reviewers, but some aspects still need to be fixed. The authors should consider each single suggestion by the second reviewer and handle it accordingly.

·

Basic reporting

The authors have responded to all the concerns and I have no further comment. I suggest accepting the revised version of this article.

Experimental design

No comments

Validity of the findings

No comments

Reviewer 3 ·

Basic reporting

The review is timely and of great interest for the community working in speech imagery in the context of EEG-based BCIs.
The Authors now justify the novelty of their review, compared to other reviews present in the literature.
The introduction is now clear.
Please, see further comments in the Additional comments section.

Experimental design

The PRISMA guidelines are followed, but the description of the inclusion and exclusion criteria needs a revision.
The sources are adequately cited, but I would like to ask checking if all the references are correctly added in the manuscript. Moreover, I have a question regarding the presence of new references compared to the previous manuscript. The question is reported in the Additional comments section.
The review is now organized in a clear manner.
Please, see further comments in the Additional comments section.

Validity of the findings

Challenges and future directions are now clearly identified and reported, following the hypothesised contribution provided in the introduction.
Please, see further comments in the Additional comments section.

Additional comments

I thank the Authors for the great effort in revising the manuscript, which has greatly improved.

There are just a few concerns I wish could be addressed, that brings me to suggest a second round of minor revisions.

The comments refer to the new clean manuscript.

- Line 31-32, please revise the end of the sentence, which I think should be “between the human brain and external devices”.
- Line 141-142, I would only say “contributions”, having that you do not pose questions but try to find something.
- The survey methodology description requires a further comment of Figure 2, considering which I have the following questions:
o First block, second column – how do you set the automation tools to say that a paper is ineligible? Moreover, what do you mean by “other reasons”?
o In the screening block, you should provide clear pointers on the reason why you have excluded some records (e.g., not in English, not on speech imagery, …).
o I am a bit confused by the eligibility block. The exclusion seems to take out EEG-based only studies, speech imagery works and works published since 2015. These seem to be the papers that requires inclusion more than exclusion, please clarify, also according to my following comment.
- Starting line 176: when describing the exclusion criteria, I think that sometimes you state inclusion criteria. Please, revise this point by providing a clear understanding of inclusion and exclusion criteria.
- Table 2, first row. Please, clarify how the “average has increased” in respect to a specific value.
- Line 535, please check how you refer to the authors.
- Figure 12, please use colours and textures that helps better identifying the corresponding labels. Please, consider also removing “DL ALL” and “ML ALL” or providing a clearer graphical representation of the fact that “DL ALL” and “ML ALL” represent the sum of the subsequent bar values.
- Considering your statement regarding the references, it is correct to have in the references, only the works you refer to. However, having that you are proposing a systematic review, it is expected that all the screen papers appear in the manuscript, even in an aggregated manner.
- Did you re-use the browsing tools to search for new papers? It seems that new papers have been presented (and according to your statements, papers published before 2015 correctly removed). Is that correct or are the new appearing papers the ones missing from the previous manuscript?

---

## Round 0.3 · accepted · Accept

The authors addressed the points raised by the reviewers and therefore I can recommend this article for acceptance.

·

Basic reporting

The authors have responded to all the concerns and I have no further comment. I suggest accepting the revised version of this article.

Experimental design

No comments

Validity of the findings

No comments

Additional comments

No comments

Reviewer 3 ·

Basic reporting

no comment

Experimental design

no comment

Validity of the findings

no comment

Additional comments

I thank the Authors for having addressed all my concerns.
I suggest the paper acceptance.